# Improving the computation efficiency of a source-oriented chemical mechanism for the simultaneous source apportionment of ozone and secondary particulate pollutants

Qixiang Xu[1, 2], Zilin Jin[1], Qi Ying[3], Ke Wang[1, 2], Fangcheng Su[1, 2*], Ruiqing Zhang[1, 2], Michael J. Kleeman[4]

1    School of Ecology and Environment, Zhengzhou University, Zhengzhou, 450001, China

2    Institute of Environmental Sciences, Zhengzhou University, Zhengzhou 450001, China

3    Zachry Department of Civil and Environmental Engineering, Texas A&M University, College Station, TX 77845-3136

4    Department of Civil and Environmental Engineering, University of California, Davis, CA 95616

**Correspondence:** Fangcheng Su (sufangcheng@zzu.edu.cn)

**Abstract:**

Source-oriented chemical mechanisms enable direct source apportionment of air pollutants by explicitly representing precursor emissions and their reaction products in atmospheric models. These mechanisms use source-tagged species to track emissions and their evolution. However, scalability was previously limited by the large number of reactions required for interactions between two tagged species, such as the NOx-NOx or VOC-NOx reactions. This study improves computational efficiency and scalability with a new method that tracks the total concentration of tagged species, reducing the $n^2$ second-order reactions for n sources into 2n pseudo first-order reactions. The overall production and removal rate of individual species remain unchanged in the new approach. The number of reactions and number of model species increase linearly with the number of source types, thus greatly improving the computation efficiency. In addition, a source-oriented Euler Backward Iterative (EBI) solver was implemented to replace the Gear solver used in previous applications of the source-oriented mechanism. The source-oriented EBI solver has been assessed by comparing predicted results with the Gear solver. Good agreement between those two methods has been achieved, as the results from the EBI scheme are linearly correlated to Gear and average of absolute relative error is below 5%. In the timing assessment, the proposed EBI scheme can effectively reduce the total chemistry time by 73%

to 90% for grids with different resolutions, which leads to the reduction of total simulation time by 46% to 74%. The proposed source-oriented scheme is efficient enough for practical long-term source apportionment applications on nested domains.

**Keywords:** Atmospheric chemical solver, Euler Backward Iterative, Source apportionment

# 1  Introduction

## 1.1  Source-oriented chemical mechanisms

Source-oriented air quality models have been used extensively in source apportionment modeling studies to determine source (or source region) contributions to $NO_x$ (Zhang & Ying, 2011), VOCs (Ying & Krishnan, 2010), secondary inorganic (Ying & Kleeman, 2006) and organic aerosols (Wang et al., 2018), and ozone (Wang et al., 2019a; Wang et al., 2020). In these models, source-tagged species and their reactions are introduced in the gas phase chemical mechanisms to track primary emissions and their reaction products from different sources. For the source apportionment of secondary aerosol products from gas-to-particle partitioning, aerosol and cloud processes are also modified to include additional model species to represent the semi-volatile products from different sources.

While this is conceptually simple, the source-oriented mechanisms are computationally expensive because the number of reactions increases almost quadratically with the number of source types due to reactions that involve two source-tagged species. For example, consider the simple reaction of $NO + NO_3 \rightarrow 2NO_2$, if the source-oriented mechanism is designed to track two explicit sources, four reactions in reaction set 1 (RS1) are needed:

$$NO^{S1} + NO_3{}^{S1} \rightarrow NO_2{}^{S1} + NO_2{}^{S1}$$
$$NO^{S1} + NO_3{}^{S2} \rightarrow NO_2{}^{S1} + NO_2{}^{S2}$$
$$NO^{S2} + NO_3{}^{S1} \rightarrow NO_2{}^{S2} + NO_2{}^{S1} \qquad \text{(RS1)}$$
$$NO^{S2} + NO_3{}^{S2} \rightarrow NO_2{}^{S2} + NO_2{}^{S2}$$

where the superscript $^{Sn}$ are tags attached to the name of the species to differentiate their source-origin. For a total of $N_s$ sources of $NO_x$ to be tracked explicitly, $N_s^2$ reactions are needed instead of one reaction. As there are quite a number of such $NO_x + NO_x$ reactions in the gas phase inorganic chemistry, the number of reactions needed for the chemical mechanism grows quickly. The necessity to deal with $N_2O_5$, which can be generated from $NO_2$ and $NO_3$ from different sources, is handled with double-source-tagged species $N_2O_{5,Sij}$.

In addition to a potential quadratic scaling of the number of reactions, the number of $N_2O_5$

species also increases quadratically with the number of explicit sources, leading to

near-quadratic growth of the overall number of species when the number of types to track

gets higher.

In ozone source apportionment calculations, it is also necessary to track the sources of

primary emitted VOCs as well as some of their reaction products in addition to the sources of

$NO_x$ (Wang et al., 2020). Some of the unsaturated VOCs such as olefins can react with the

$NO_3$ radical. In the source-oriented mechanism, the number of reactions needed for these

VOC+$NO_3$ reactions also increases quadratically, as shown in RS2 below, using the ethene

(ETHE) + $NO_3$ reaction from the SAPRC-07 mechanism (Carter, 2010) as an example for

two sources. For accurate VOC source apportionment calculations that involve reactions

between two source-oriented species, such quadratic dependence of source types and

reaction numbers also arises (Ying & Krishnan, 2010).

$$TERP^{S1} + NO_3^{S1} \rightarrow TRPRXN^{S1} + 0.287xNO_2^{S1} + 1.786RO2\_R + ...$$
$$TERP^{S1} + NO_3^{S2} \rightarrow TRPRXN^{S1} + 0.287xNO_2^{S2} + 1.786RO2\_R + ...$$
$$TERP^{S2} + NO_3^{S1} \rightarrow TRPRXN^{S2} + 0.287xNO_2^{S1} + 1.786RO2\_R + ...$$
$$TERP^{S2} + NO_3^{S2} \rightarrow TRPRXN^{S2} + 0.287xNO_2^{S2} + 1.786RO2\_R + ...$$

(RS2)

Due to the necessity of explicitly handling some or all of these reactions in

source-oriented mechanisms, the source-oriented modeling approach is computationally

intensive so that previous applications were limited to up to 9 explicit sources for secondary

nitrate in a single run (Kleeman & Cass, 2001; Ying et al., 2004; Ying et al., 2014). In some

previous work for VOC and secondary organic aerosol source apportionment, only one

explicit source was tracked at a time to simplify the reactions and to reduce the computation

burden (Ying & Krishnan, 2010; Wang et al., 2018). However, multiple model runs are

needed to determine the contributions from all sources. To make the source-oriented

approach practical for a larger number of source types, it is necessary to improve the

computation efficiency of the source-oriented approach.

**1.2 Numerical solution of stiff ODEs for gas phase reaction kinetics**

The gas phase chemical reaction kinetics are described by of a non-linear system of stiff

ordinary differential equations (ODEs) which must be solved to predict the transient

evolution of the concentrations of gas species. One of the most widely used schemes is the

Gear method (Ralph, 1973)

$$(\boldsymbol{I} - \hbar\beta_s\boldsymbol{J})\Delta\vec{C}_t^{m+1} = -\vec{C}_t^m + \sum_{j=1}^{s}\alpha_j\vec{C}_{t-jh} + \hbar\beta_s\frac{d\vec{C}_t^m}{dt} \qquad (1)$$

where $h$ is the integration time step; $\vec{C}_t^m$ is the vector of species concentrations to be solved for time t during the $m^{th}$ iteration; $\Delta\vec{C}_t^{m+1}$ is the correction vector to estimate $\vec{C}_t^{m+1}$; $\boldsymbol{J}$ is the Jacobian matrix of partial derivatives for all species that $J_{i,k} = \frac{\partial^2 C_{i,t}}{\partial C_{k,t}\,\partial t}$. It is calculated either analytically or numerically initially based on $\vec{C}_{t-h}$ and updated when necessary using the most recent values of $\vec{C}_t$; $\boldsymbol{I}$ is the identity matrix; $s$ is the order of the accuracy; $\beta_s$ and $\alpha_j$ are scalar multipliers that depend on the order of the method. For each iteration, the new concentrations for the next time step t is evaluated as $\vec{C}_t^{m+1} = \vec{C}_t^m + \Delta\vec{C}_t^{m+1}$. The iteration stops when $\Delta\vec{C}_t^{m+1}$ becomes less than a provided error. A practical solver also needs to automatically adjust the time step size $h$, the order of accuracy $s$ and recalculate the Jacobian matrix when necessary to ensure that the estimated error in one time step is less than a prescribed criteria (Jacobson, 2006).

The advantage of using the Gear solver is that it is a general stiff solver so that no special modifications are needed for a specific chemical mechanism. However, it is computationally intensive as it involves evaluating the Jacobian matrix and performing LU factorization for the left-hand side matrix. A Sparse-Matrix Vectorized Gear (SMVGEAR) solver was developed by Jacobson and Turco (1994) and has been included in a number of atmospheric chemical transport models (Zhang et al., 2011; Hu et al., 2012; Hu et al., 2014). The SMVGEAR solver was also used previously to solve the gas phase reaction kinetics in the source oriented CTM (Shi et al., 2017; Li et al., 2021). Test based on the reported Texas Air Quality Study 2006 ozone episode showed that a source-oriented SAPRC-07 mechanism that simultaneously performs the source apportionment of $NO_x$, $SO_2$, primary VOCs, HCHO, and ozone for 16 sources needs 11 times of the computation time of the original non-source-oriented mechanism (Parrish et al., 2009). The gas chemistry is the most time-consuming step that normally takes more than half of total simulation time, as shown in Figure 1.

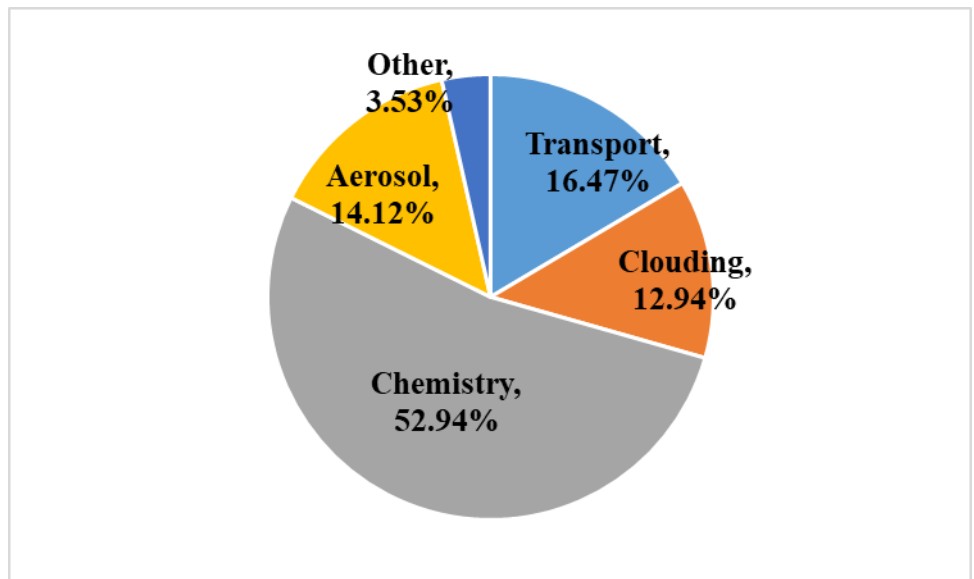

**Fig. 1.** Typical fraction of time spent in the scientific processes in the source-oriented CMAQ model. This is based on a 36-km resolution domain (160 rows*130 columns*44 layers), 6 source types and solved using the SMVGEAR solver.

The Euler Backward Iterative (EBI) (Hertel et al., 1993) is a faster method to solve the stiff ODE systems arising from a gas-phase photochemical mechanism. The basis of this method is the backward Euler method as shown in equation (2),

$$C_{i,t} = \frac{C_{i,t-h} + hP_{i,t}}{1 + hL_{i,t}} \tag{2}$$

where $C_{i,t}$ and $C_{i,t-h}$ are the concentrations of species i at time t and t-h, respectively; h is the integration time step; $P_{i,t}$ and $L_{i,t}$ are the chemical production and loss terms, respectively, evaluated using the concentrations of the species at time t. Equation (2) represents a set of coupled non-linear equations and a solution can be obtained by first evaluating the production and loss terms using the concentrations from the previous time step *t-h* to calculate an initial estimation of the species concentrations for the time step *t* based on equation (2). These concentrations are applied to update the *P* and *L* terms so that and updated estimation of the species concentrations for time step *t* are obtained. This procedure is repeated until the changes in $C_{i,t}$ for a new iteration are less than a prescribed value.

For atmospheric photochemical reactions, there are several families of species whose concentrations are strongly coupled in reversible reactions. The general backward Euler method described above has a slow rate of convergence or even fails to converge. In the EBI solver, four family groups of strongly coupled species, are excluded from the general equation (2): (1) NO, $NO_2$, $O_3$ and $O(^3P)$, (2) OH, $HO_2$, HONO and $HNO_4$, (3) peroxyacetyl

radical (CH3C(O)OO·, or $C_2O_3$) and peroxyacetyl nitrate (PAN), and (4) $NO_3$ and $N_2O_5$. For these species, analytical solutions of four sets of non-linear algebraic equations (Eqs. 9-12 in Hertel et al., 1993) are applied to determine their concentrations at time t instead of using the P and L terms with equation (2). The detailed mathematical procedures involved in obtaining these analytical solutions are listed in Appendices A.1-A.4 of Hertel et al. (1993).

The accuracy of the EBI solver has been evaluated against more accurate solvers (Hertel et al., 1993) and is the default choice in the CMAQ model for a number of built-in chemical mechanisms. However, it cannot be directly used to solve the source-oriented chemical mechanisms since the solution procedures of aforementioned four strongly coupled groups only calculate the total concentrations of species, and source-tagged species are not included in the explicit solutions. Thus, for source-oriented mechanisms, groups of tagged reactive nitrogen species require additional solution steps. Direct replication of the procedures for total concentrations to treat the source tagged species are infeasible, as this would dramatically increase the computational cost and the difficulty of source code implementation. An EBI solver capable of handling the chemical mechanisms with source tagged species and their reactions while maintaining brevity and ease of implementation, is highly desirable. Ideally, it should be able to predict the concentrations of source-tagged reactive nitrogen species based on their pre-determined total concentrations, which would greatly improve the computational efficiency and consequently enhance the applicability of the source-oriented air quality model.

The objective of this study is to develop a computationally efficient source-oriented gas phase chemical mechanism for the simultaneous source apportionment of $O_3$ and other gaseous pollutants such as CO, primary VOCs, NO, $NO_2$, $SO_2$ and $NH_3$. The mechanism, when linked with a proper source-oriented aerosol mechanism, can be used to determine the sources contributions to nitrate, sulfate and ammonium ion. The method for improving the efficiency of the source-oriented mechanism through simplification of reaction representation and modification of the EBI ODE solver for source-oriented nitrogen species is described in Section 2. Section 3 details the testing of the improved mechanism and the source-oriented EBI solver.

## 2 Method

## 2.1 Reduce the number of reactions and source-tagged species

In the original source-oriented model, a general reaction set that involves two source-tagged species as reactants for $N_s$ source types can be written in the following form:

$$A_i + B_j \xrightarrow{k_{AB}} C_i + D_j + E, \quad i = 1,2,\ldots,N_s; j=1,2,\ldots,N_s \qquad \text{(RS3)}$$

where $A$, $B$, $C$ and $D$ are source-tagged model species and the subscripts denote source origin index of these species. For simplicity, assuming that $C$ and $D$ are the reaction products from $A$ and $B$, respectively. $E$ represents a general product whose source-origin is not tracked in the model simulation. $k_{AB}$ is the second order reaction rate coefficient, which is the same for all the reactions in this reaction set. Reaction sets RS1 and RS2, as shown in the examples in Section 1, can both be expressed in this form. The loss rate of $A_i$ is calculated using equations (3a) and (3b):

$$\frac{d[A_i]}{dt} = -k_{AB}[B_1][A_i] \ldots - k_{AB}[B_{N_s}][A_i] = -k_{A,effB}[A_i] \qquad \text{(3a)}$$

$$k_{A,effB} = k_{AB}[B_{tot}] = k_{AB}\sum_{j=1}^{N_s}[B_i] \qquad \text{(3b)}$$

where $k_{A,effB}$ is the pseudo-first order reaction rate coefficient for $[A_i]$ based on the total concentration of $B$, $[B_{tot}]$, as defined in equation (3b). This method is valid for reactions with rate laws of the form rate=$k[A]^m[B]^n$..., where the rate is proportional to the product of reactant concentrations raised to their orders. A similar set of equations can be derived for the loss rate of individual tagged species $B_i$. Thus, the $N_s^2$ second-order reactions represented by RS3 can be equivalently described by the following $2N_S$ pseudo first-order reactions,

$$A_i \xrightarrow{k_{A,effB}} C_i + E, \quad i=1,2,\ldots,N_s$$
$$B_i \xrightarrow{k_{B,effA}} D_i, \qquad i=1,2,\ldots,N_s \qquad \text{(RS4)}$$

For the non-typed product E, it can appear in either the $A_i$ reactions or the $B_i$ reactions and it is easy to show that the formation rate of $E$ based on RS4 is the same as these from RS3.

The double-tagged $N_2O_5$ species and their reactions can be simplified as well. For $N_2O_{5,ij}$ which represents $N_2O_5$ based on $NO_2$ from source $i$ and $NO_3$ from source $j$, it can be equivalently written as $0.5N_2O_{5,i} + 0.5N_2O_{5,j}$ as shown in reaction set RS5, in terms of preserving the source contributions to $NO_2$ and $NO_3$,

$$NO_{2,i} + NO_{3,j} \xrightarrow{k_{NO2NO3}} 0.5N_2O_{5,i} + 0.5N_2O_{5,j} \quad , i=1,2,\ldots,N_s; j=1,2,\ldots,N_s \qquad \text{(RS5)}$$

With this simplification, as well as the pseudo-first order reaction technique described above, the reactions of $N_2O_5$ formation from $N_s$ types of $NO_x$ can be expanded into $2N_s$ reactions with $N_s$ tagged $N_2O_5$ species as shown in the general reaction set RS4, where $A$ is $NO_2$, $B$ is $NO_3$ and $C$ and $D$ are $N_2O_5$ species that have the same source tag as A and B, respectively. This dual-tagged reaction reduction method significantly decreases the number of species as well as the number of reactions for the source-oriented mechanism.

The total concentration of tagged species needed in equation (3b) for the pseudo first order reaction rate coefficient need not be tracked separately in the dual-tagged reaction reduction method. Instead, they are calculated on-the-fly and then used to calculate the pseudo first-order rate coefficients for the reactions of the tagged species shown above. The function that calculates the reaction rates to be used in the stiff ODE solvers needs to be modified to recognize these special pseudo first-order reactions. The CMAQ model is capable of dealing with these special pseudo-first order reaction natively with its included mechanism preprocessor (CHEMMECH). An example input to the CHEMMECH on how the reactions are constructed for $NO + NO_3 \rightarrow 2NO_2$ for 10 source types is illustrated in List 1 of Appendix.

## 2.2 Source-oriented Euler Backward Iteration (EBI) scheme

### 2.2.1 Solution for family of source-tagged species in coupled reversible reactions

For source apportionment of ozone and secondary inorganic aerosols, the reactive nitrogen species NO, $NO_2$, HONO, $HNO_4$, PAN, $NO_3$ and $N_2O_5$ are source-tagged. For PAN, its source is determined by the source of $NO_2$ while the peroxyacetyl group is not source-tagged.

The standard solution procedure of the source-tagged species in the source-oriented EBI method includes two major steps: (1) evaluating of total concentrations of these tagged species using equation sets (9), (10) and (11) and (12) in Hertel et al. (1993); and (2) predicting the concentrations for each tagged species based on the total concentrations. In the following, equations for step (1) are summarized first, followed by equations for step (2). Equations are separately listed for each family.

The first set of equations (4a-4d) are used to solve the total concentrations of NO, $NO_2$, $O_3$ and $O(^3P)$. These equations are based on the corresponding ones in Hertel et al. (1993).

$$[NO_{tot}]_t = [NO_{tot}]_{t-h} + hP_1' + h(r_{1,2} + J_1)[NO_{2,tot}]_t - hk_{1,3}[O_3]_t[NO_{tot}]_t$$
$$- hr_{2,1}[NO_{tot}]_t - hL_1'[NO_{tot}]_t \tag{4a}$$

$$[NO_{2,tot}]_t = [NO_{2,tot}]_{t-h} + hP_2' + hr_{2,1}[NO_{tot}]_t + hk_{1,3}[O_3]_t[NO_{tot}]_t$$
$$- h(r_{1,2} + J_1)[NO_{2,tot}]_t - hL_2'[NO_{2,tot}]_t \tag{4b}$$

$$[O_3]_t = [O_3]_{t-h} + hJ_2[O(^3P)]_t - hk_{1,3}[NO_{tot}]_t[O_3]_t - hL_3'[O_3]_t \tag{4c}$$

$$[O(^3P)]_t = [O(^3P)]_{t-h} + hP_{12}' + hJ_1[NO_{2,tot}]_t - hJ_2[O(^3P)]_t - hL_{12}'[O(^3P)]_t \tag{4d}$$

217 In the above equations, species with a subscript *tot* represent the total concentration of a
218 set of tagged species, which is calculated by adding the concentrations of the individual
219 tagged species; $h$ is the size of the current time step; $r_{1,2}$ is the production rate coefficient of
220 NO from $NO_2$, excluding photo-dissociation and $J_1$ is the photolysis rate of $NO_2$ to form
221 NO; $k_{1,3}$ is the second-order rate coefficient for $NO+O_3$ reaction to form $NO_2$; $r_{2,1}$ is the
222 pseudo-fist order rate constant for the production of $NO_2$ from NO from all other pathways,
223 excluding the $NO+O_3$ reaction. $J_2$ is the first-order reaction rate constant for $O(^3P) + O_2$ to
224 form $O_3$. The terms $P_1'$, $P_2'$, and $P_{12}'$ account for all the remaining production for the total
225 concentrations of NO, $NO_2$ and $O(^3P)$ in the mechanism, and the terms $L_1'$, $L_2'$, , $L_3'$, and $L_{12}'$
226 are the losses of the total concentrations of NO, $NO_2$, $O_3$ and $O(^3P)$, respectively. Analytical
227 solutions for the total concentrations based on 4a-4d were derived and described in detail in
228 the Appendix A of Hertel et al. (1993) and are not repeated here.

229 Once the total concentrations of NO, $NO_2$, $O_3$ and $O(^3P)$ are solved, concentrations of the
230 source-tagged NO and $NO_2$ are solved from the following two equations,

$$[NO_i]_t = [NO_i]_{t-h} + hP_{1,i}' + h(r_{1,2} + J_1)[NO_{2,i}]_t - hk_{1,3}[O_3]_t[NO_i]_t - hr_{2,1}[NO_i]_t$$
$$- hL_1'[NO_i]_t \tag{5a}$$

$$[NO_{2,i}]_t = [NO_{2,i}]_{t-h} + hP_{2,i}' + hr_{2,1}[NO_i]_t + hk_{1,3}[O_3]_t[NO_i]_t - h(r_{1,2} + J_1)[NO_{2,i}]_t$$
$$- hL_2'[NO_{2,i}]_t \tag{5b}$$

231 where i is the source index for the tagged species. For each i, the two unknowns $[NO_i]_t$
232 and $[NO_{2,i}]_t$ are solved analytically using the following equations,

$$[NO_i]_t = \frac{1}{\det(A)}(A_{22}b_1 - A_{12}b_2) \tag{6a}$$

$$[NO_{2,i}]_t = \frac{1}{\det(A)}(-A_{21}b_1 + A_{11}b_2) \tag{6b}$$

$$A = \begin{bmatrix} 1 + hk_{1,3}[O_3]_t + hr_{1,2} + hL'_1 & -h(r_{2,1} + J_1) \\ -h(r_{2,1} + k_{1,3}[O_3]_t) & 1 + hr_{1,2} + hJ_1 + hL'_2 \end{bmatrix} \tag{6c}$$

$$b = \begin{bmatrix} [NO_i]_{t-h} + hP'_{1,i} \\ [NO_{2,i}]_{t-h} + hP'_{2,i} \end{bmatrix} \tag{6d}$$

In equations 6a and 6b, $\det(A)$ is the determinant of the 2 x 2 matrix $A$, as defined in equation 6c.

The second set of equations are for the total concentrations of OH, HO$_2$, HONO and HNO$_4$, as shown in equations (7a) - (7d),

$$[OH]_t = [OH]_{t-h} + hP'_4 + hr_{4,5}[HO_2]_t + hr_{4,19}[HONO_{tot}]_t - hL_4[OH]_t \tag{7a}$$

$$[HO_2]_t = [HO_2]_{t-h} + hP'_5 + hr_{5,4}[OH]_t + hr_{5,21}[HNO_{4,tot}]_t - 2hk_{5,5}[HO_2]_t^2 \\ - hL'_5[HO_2]_t \tag{7b}$$

$$[HONO_{tot}]_t = [HONO_{tot}]_{t-h} + hr_{19,4}[OH]_t - hL_{19}[HONO_{tot}]_t \tag{7c}$$

$$[HNO_{4,tot}]_t = [HNO_{4,tot}]_{t-h} + hr_{21,5}[HO_2]_t - hL_{21}[HNO_{4,tot}]_t \tag{7d}$$

The notations and symbols used in the above equations are similar to those used in equations (5a-5d). $r_{4,5}$ and $r_{4,19}$ are pseudo first order production rate coefficients of OH from HO$_2$ and HONO, respectively. $r_{5,4}$ and $r_{5,21}$ are pseudo first order production rate coefficients of HO$_2$ from OH and HNO$_4$, respectively. $k_{5,5}$ is the HO$_2$+HO$_2$ self-reaction rate coefficient. $r_{19,4}$ and $r_{21,5}$ are pseudo first order rate coefficients for the production of HONO and HNO$_4$ from OH+NO and HO$_2$+NO$_2$, respectively. The terms $P'_4$ and $P'_5$ account for all the remaining production of OH and HO$_2$ and the terms $L_4$, $L'_5$, $L_{19}$ and $L_{21}$ account for all the other losses of OH, HO$_2$, HONO and HNO$_4$, respectively. Analytical solutions for (7a)-(7d) were also derived and described in detail in Appendix A of Hertel et al. (1993). As the OH and HO$_2$ concentrations are determined, concentrations of individual HONO and HNO$_4$ from different sources are solved using equations 8a and 8b, respectively.

$$[HONO_i]_t = \frac{[HONO_i]_{t-h} + hr^i_{19,4}[OH]_t}{1 + hL_{19}} \tag{8a}$$

$$[HNO_{4,i}]_t = \frac{[HNO_{4,i}]_{t-h} + hr^i_{21,5}[HO_2]_t}{1 + hL_{21}} \tag{8b}$$

where $r^i_{19,4}$ and $r^i_{21,5}$ are pseudo first order production rate coefficients of HONO and HNO$_4$ from source i due to NO and NO$_2$ from the same source with OH and HO$_2$, respectively. Concentrations of NO$_i$ and NO$_{2,i}$ for the current timestep has already been

determined using (6a) and (6b) and these concentrations will be applied to calculate $r^i_{19,4}$ and $r^i_{21,5}$ used in the above two equations.

The third set of equations are for $C_2O_3$ and PAN:

$$[C_2O_3]_t = [C_2O_3]_{t-h} + hP'_8 + hr_{8,9}[PAN_{tot}]_t - 2hk_{8,8}[C_2O_3]_t^2 - hL'_8[C_2O_3]_t \tag{9a}$$

$$[PAN_{tot}]_t = [PAN_{tot}]_{t-h} + hr_{9,8}[C_2O_3]_t - hL_9[PAN_{tot}]_t \tag{9b}$$

In the equation, it is assumed that in the source-oriented mechanism, PAN is a source-tagged species and its source is based on the source of $NO_2$ but $C_2O_3$ is not source-tagged. This is sufficient for the source apportionment of ozone, as described in the Section 2.1. A quadratic equation for $C_2O_3$ can be obtained (see Appendix A of Hertel et al. 1993). In $r_{9,8}$, total $NO_2$ concentration at the current timestep t has already been determined by solving equations 5a-5d. Once the concentrations of $C_2O_3$ at timestep t is solved, the concentrations of each of the tagged PAN species can be used by:

$$[PAN_i]_t = \frac{[PAN_i]_{t-h} + hr^i_{9,8}[C_2O_3]_t}{1 + hL_{21}} \tag{9c}$$

where the $r^i_{9,8}$ includes the concentration of $NO_{2,i}$ ($NO_2$ attributed to source i) at the current timestep t.

The last set of equations treated specially in the source-oriented EBI solver is for $NO_3$ and $N_2O_5$, as shown in equations (10),

$$[NO_{3,tot}]_t = [NO_{3,tot}]_{t-h} + hP'_{15} + hr_{15,16}[N_2O_{5,tot}]_t - hL_{15}[NO_{3,tot}]_t \tag{10a}$$

$$[N_2O_{5,tot}]_t = [N_2O_{5,tot}]_{t-h} + hr_{16,15}[NO_{3,tot}]_t - hL_{16}[N_2O_{5,tot}]_t \tag{10b}$$

The two equations are linear equations and can be solved easily for $NO_{3,tot}$ and $N_2O_{5,tot}$. The $NO_{3,i}$ and $N_2O_{5,i}$ (as discussed in section 2.1, sources of $N_2O_5$ can be tracked with a single type instead of double typed) can be solved explicitly based on the two equations as well, as shown in (10c) and (10d),

$$[NO_{3,i}]_t = \frac{(1 + hL_{16})\left([NO_{3,i}]_{t-h} + hP^{i,'}_{15}\right) + hr_{15,16}[N_2O_{5,tot}]_{t-h}}{(1 + hL_{15})(1 + hL_{16}) - h^2 r_{15,16} r_{16,15}} \tag{10c}$$

$$[N_2O_{5,i}]_t = \frac{(1 + hL_{15})[N_2O_{5,tot}]_{t-h} + hr_{16,15}\left([NO_{3,i}]_{t-h} + hP^{i,'}_{15}\right)}{(1 + hL_{15})(1 + hL_{16}) - h^2 r_{15,16} r_{16,15}} \tag{10d}$$

Note that $r_{16,15}$ includes the total concentration of $NO_2$ thus is the same as that used in equation (10b). However, the production of $NO_3$ from other reactions do have to be source specific, thus the P15's used in equation (10c) and (10a) are different.

To avoid round-off errors introduced in the calculation for the source-tagged species involved in these four groups so that the sum of the source-tagged species always exactly matches the total concentrations, their concentrations are readjusted by the pre-determined total concentrations of photochemical species by solving the algebraic equations for the special groups in original EBI scheme.

### 2.2.2 Successive under-relaxation

During the testing of the above algorithm, non-converging oscillations were sometimes observed, mostly due to the low concentration source-tagged species. The iterative process used in the EBI solver can be modified to include a relaxation factor $\alpha$ so that the concentration array at end of each iteration is updated by a weighted average of the results from the previous iteration and the present estimated values in the current iteration, as shown in equation (11),

$$C_{i,iter}^{update} = (1 - \alpha)C_{i,iter-1} + \alpha C_{i,iter} \tag{11}$$

The selection of $\alpha$ influences only the number of iterations required for convergence, not the final converged solutions. Generally, larger $\alpha$ values lead to faster convergence, but have a higher chance of fall into oscillation. Based on the testing, $\alpha=0.8$ appears to be a conservative choice that always lead to convergence. A dynamic under-relaxation scheme using a set of varying $\alpha$ values between 0.79 and 1.0, based on the number of iterations in the EBI scheme, is shown to lead to faster convergence. This is further discussed in the Results section.

### 2.3 Test mechanism and model set up

To evaluate how much improvement in computation efficiency can be achieved by using the simplified reaction representation and source-oriented EBI solver, a series of source-oriented mechanisms for simultaneous attribution of ozone and secondary inorganic aerosol were constructed based on the SAPRC-07 photochemical mechanism and implemented in CMAQv5.2. The SAPRC-07 mechanism is chosen instead of the more recent versions of SAPRC because it is faster with fewer species and reactions, and thus is more suitable for simulations requiring rapid response, such as operational air quality forecasting and for source apportionment of ozone and secondary inorganic aerosols. The source-oriented mechanism based on this will be applied in a future air quality forecasting model that also forecasts source-tagged species concentrations and source-region

contributions to air pollution. As the primary goal of this paper is to evaluate the efficiency

of the gas phase algorithm, aerosol results despite being enabled in the simulations along

with cloud processes, are not included in the analyses described below.

The tested SAPRC-07 mechanism used in this study contains a total of 134 species and

341 reactions. Among these species, 15 species are reactive nitrogen species. For each of

these species, tagged species are used to track their source origins. Reactions involving these

species are expanded in the source-oriented mechanism. In addition, CO, $SO_2$ and sulfuric

acid (SULF) were also expanded in the source-oriented mechanism. To evaluate source

contributions to ozone, 14 primary VOC species were also treated as source-oriented species

in addition to source-tagged non-reactive $O_3$ species to track contributions from different

sources of NOx and VOCs to $O_3$ formation. As HCHO is an important oxidation product

from several parent VOCs, sources of secondary HCHO from the first-generation oxidation

of parent VOCs are also tracked. Details for the source apportionment of $O_3$ has been

described by Wang et al. (Wang et al., 2019a; Wang et al., 2019b) and are not repeated here.

While the dual-tagged reaction reduction method and EBI solver for source apportionment

can be employed separately, their integration could lead to enhanced performance and more

significant benefits. Two versions of the source-oriented SAPRC-07 mechanisms are

prepared. The first version (V1) uses double-tagged $N_2O_5$ and fully expanded reactions

without the pseudo first order reactions described in Section 2. The second version (V2) is

based on single-tagged $N_2O_5$ and a dual-tagged reaction reduction treatment applied to fully

expanded V1 mechanism, as descried in section 2.1. Both mechanisms are constructed to

track emissions from ten source types. The number of reactions and species in each

mechanism is listed in Table 1, while the accuracy of this method is presented in the Fig A1

which shows the EBI solver's predictions scattered on the 1:1 line when compared to the

SMVGEAR solver's results for of tagged species concentrations and the total.

**Table 1.** Computation time needed for a one-day simulation using two different versions of the source-oriented chemical mechanism and two versions of the ODE solvers. Both versions are capable of tracking 10 different source types in a single simulation.

| | V0 SMVGEAR | V1 SMVGEAR | V2 SMVGEAR | | | V2 EBI* | | |
|---|---|---|---|---|---|---|---|---|
| Domain resolution | 36km | 36km | 36km | 12km | 4km | 36km | 12km | 4km |
| # of total species | 134 | 512 | 422 | | | | | |

| | | | | | | | | |
|---|---|---|---|---|---|---|---|---|
| # of total reactions | 341 | 2845 | 1376 | | | | | |
| Total Chem. Time^ (min) | 15.3 | 133 | 49.9 | 172 | 325 | 13.4 | 15.1 | 32.4 |
| Total Sim. Time^ (hr) | 0.419 | 2.63 | 1.32 | 3.59 | 6.27 | 0.707 | 1.03 | 1.63 |
| Chem % | 61% | 84% | 63% | 80% | 86% | 32% | 24% | 33% |
| Chem time reduction wrt. V1 | 12% | - | 62% | - | - | 90% | - | - |
| Chem time reduction wrt. V2-S[#] | 31% | - | - | - | - | 73% | 91% | 90% |
| Total time reduction wrt. V1 | 16% | - | 50% | - | - | 73% | - | - |
| Total time reduction wrt. V2-S[#] | 32% | - | - | - | - | 46% | 71% | 74% |

* Simulation time based on dynamic under-relaxation coefficient.

^ Wall-clock time

# V2-S: V2 reaction mechanism with SMVGEAR solver

The testing is performed for totally threedays (July 1-3, 2020) simulation using three-level nested domains with 36, 12 and 4 km resolutions that cover eastern Asia, central and eastern China and Henan province in central China, respectively. The meteorology inputs were based on the Weather Research and Forecasting (WRF) model v4.1.4. The anthropogenic emissions are based on the Multi-scale Emission Inventory for China (MEIC, for 36- and 12-km domains, available from http://www.meicmodel.org/) and a local emission inventory (for the 4-km domain) (Lu et al., 2023). Emissions are grouped into six source categories, including five anthropogenic source sectors (power, industrial, residential, transportation and agriculture) and one biogenic sector, whose emission is based on MEGAN (Model for Emissions of Gases and Aerosol from Nature) v2.1(Guenther et al., 2006). The initial concentrations of the species are based on a 7-day non-source-oriented simulation. The boundary conditions for the 36-km domain are based on the clean continental vertical profiles included in the CMAQ model. The boundary conditions of the 12- and 4-km domains are based on results from the parent 36- and 12-km domains, respectively.

Three sets of simulations using the source-oriented mechanisms were conducted: (1) V1, solved using the SMVGEAR solver (V1-GEAR), (2) V2, solved with SMVGEAR (V2-GEAR), and (3) V2, solved with source-oriented EBI (V2-EBI). For the SMVGEAR a relative tolerance (RTOL) of $1x10^{-3}$ and an absolute tolerance (ATOL) of $1x10^{-9}$ were used. For the EBI solver, only a relative tolerance RTOL was used in the convergence check. For most of the species, a relative tolerance of $1x10^{-3}$ is used. Exceptions include pseudo-steady-state species like $O_3P$ and $O_1D$, for which a tolerance of 1.0 is applied, indicating that a convergence check is not applicable. Integrate reaction rate analysis (IRR) was used in all three simulations as it is needed for the ozone source apportionment algorithm (Wang et al., 2019b). In addition to the source-oriented model simulations, two

base case simulations (V0) were conducted using the unmodified SAPRC-07 solved with the
SMVGEAR (V0-SMVGEAR) and the EBI (V0-EBI) solvers, respectively.

All the simulations were conducted on a Dell Precision-Tower 7810 working station
(2XE-2660-v4, 28/56 cores/threads and 256G of DDR4 RAM), and the run was in parallel in
a configuration of 8x6 domain decomposition.

## 3 Results

### 3.1 Timing results

For the first day simulation, the wall-clock time for the gas-phase mechanism as well as
the total run time were recorded using the time function (MPI_WTIME) in the Massage
Passing Interface. An MPI_BARRIER call was issued before each MPI_TIME call to make
sure that the measured wall-clock time represents the actual time for a chemistry time step in
a static domain decomposition setting with imbalanced loads.

The wall-clock time for the 1-day simulation using the source-oriented EBI solver was
compared with the SMVGEAR solver and presented in Table 1. The fully expanded
source-oriented mechanism with SMVGEAR (V1-SMVGEAR) is the slowest and only the
36-km resolution simulation was conducted. The simplified reaction representation alone
(V2-SMVGEAR, see section 2.1) leads to a reduction of chemistry time by ~62% when
compared with V1-SMVGEAR (133 min to 49.9 min) and total computation time by ~50%
(2.63 hr to 1.32 hr). Using the source-oriented EBI on V2 (V2-EBI) further reduces the total
chemistry time to ~13.4 min. Compared to the V1-SMVGEAR, V2-EBI reduced the
chemistry time by ~90% and the total time by ~75% for a one-day simulation in the 36 km
domain.

The EBI solver represents a significant reduction in both chemistry time and total
computation time comparing to the SMVGEAR solver. When V2-SMVGEAR and V2-EBI
are compared, the total chemistry time saving of EBI scheme increases with grid resolution,
from 70% for 36 km to 90% for 4 km grid. As the result, the total simulation time saving
increases from 46% for coarse grid to 74% for fine grid. The increase in time saving of EBI
solver with grid resolution can be attributed to the smaller time step size which is determined
from flow Courant stability criterion. For fixed simulation duration (1 day in this study), the
required total number of chemistry steps increase dramatically with grid resolution, this
results in significant reduction in total chemistry time with faster EBI scheme for finer grid.

Therefore, for time consuming applications such as long-term source apportionment

simulation of nested domains, the time efficiency can be improved by a factor of 3 or more.

The V2-EBI results shown in Table 1 are based on the dynamic under-relaxation using an

iteration count dependent under-relaxation factor (α) as shown in Table 2. In this scheme, the

α is initially set to 1.0 and gradually decreases to smaller values. If the solution does not

converge in 15 iterations, a constant α of 0.79 is used. Using this dynamic α scheme is

demonstrated to be more efficient than using a constant α, as shown in Figure 2. The optimal

value for fixed α is 0.8, at which the total chemistry time could be reduced by ~62%, for the

36-km domain simulations, which is approximately 10% less than the dynamic α scheme.

The values of α in Table 2 have been fine-tuned through a series of numerical

experiments to optimize the convergence rate for the source-oriented chemical mechanisms.

The dynamic α strategy demonstrates superior performance due to its adaptive nature. Larger

α values in the early iterations aggressively propel a rapid movement of the solution vector

towards the region of the true solution. Subsequently, in later iterations, a more conservative

approach with smaller α values is adopted to gradually refine the solution and effectively

damp out residual errors and potential overshoots by assigning more weight to the solution

of the previous iteration. For the majority of test cases, convergence is achieved within 10

iterations. However, in instances where convergence is slower, a slightly larger adjustment

step (an increase in α between iterations 11 and 15) can be beneficial for a faster approach to

the true solution. The ultimate value of α=0.79 is aimed at effectively damping oscillations in

the final convergence stages.

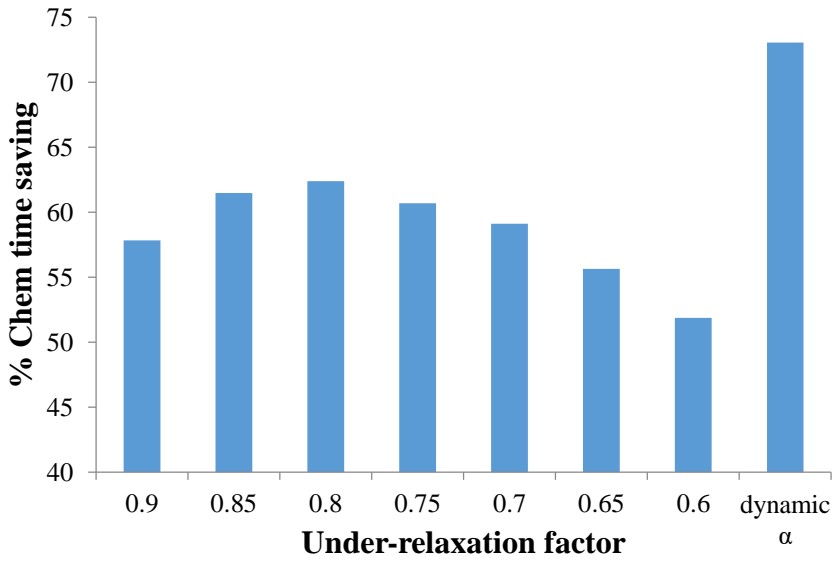

**Fig. 2**. Percentage chemistry time saving with respect to the SMVGEAR solver for the

identical source-oriented chemical mechanism for the 36-km domain. See text and Table 2

for the details of the dynamic under-relaxation scheme.

**Table 2.** Dynamic section of the under-relaxation factor ($\alpha$) based on iteration count

| Iteration # | $\alpha$ | Iteration # | $\alpha$ |
|---|---|---|---|
| 1 | 1.00 | 6-7 | 0.79 |
| 2 | 0.90 | 8-10 | 0.78 |
| 3 | 0.85 | 11-15 | 0.80 |
| 4-5 | 0.81 | $\geq$16 | 0.79 |

## 3.2 Accuracy assessment of the source oriented EBI solver

The general accuracy of the EBI solver has been tested by Hertel et al. (1993). The

accuracy of proposed source apportionment EBI scheme was evaluated by comparing

predicted results with those from the SMVGEAR. For this comparison, results from the first

420 day of a three-day simulation are primarily presented to highlight the maximum potential

discrepancy between EBI and SMVGEAR.

Hourly average concentrations of the three days NO, $NO_2$, PAN, HONO, OH and $HO_2$ at

each grid cell in the surface layer of the 36-km domain were selected as sample species for

accuracy evaluation because these are the important species treated specially by the source

oriented EBI solver. The maximum and mean values of the normalized error of these species

are listed in the Table 3. For all these species, the maximum normalized error among all grid

cells is less than 15% and the mean normalized error does not exceed 4% on the first day.

Subsequently, the error gradually decays over the following two days, reaching an order of

magnitude of 0.1% to 1% by the third day. This indicates that the accuracy of the

source-oriented EBI scheme is acceptable, as the errors are anticipated to diminish further

with increasing flow time.

**Table 3.** Max and mean values of the normalized error* for selected species in the 36-km

domain.

| Species | Max Normalized error (%) | | | Mean Normalized Error (%) | | |
|---|---|---|---|---|---|---|
| | Day1 | Day2 | Day3 | Day1 | Day2 | Day3 |
| NO_X0 | 14.45 | 8.83 | 6.49 | 3.01 | 2.41 | 1.14 |
| NO_X1 | 5.92 | 3.49 | 1.85 | 2.50 | 1.41 | 1.05 |
| NO_X2 | 2.30 | 1.48 | 1.05 | 0.0059 | 0.0034 | 0.0025 |
| NO_X3 | 6.29 | 3.62 | 1.70 | 1.35 | 0.72 | 0.45 |
| NO_X4 | 5.64 | 2.00 | 1.33 | 0.59 | 0.36 | 0.19 |
| NO_X5 | 0.99 | 0.40 | 0.24 | 0.57 | 0.31 | 0.17 |

| | | | | | | |
|---|---|---|---|---|---|---|
| NO_X6 | 1.35 | 0.65 | 0.32 | 0.41 | 0.24 | 0.15 |
| $NO_2$_X0 | 12.69 | 9.16 | 6.49 | 3.99 | 2.34 | 1.26 |
| $NO_2$_X1 | 11.89 | 7.06 | 5.61 | 3.37 | 1.19 | 0.86 |
| $NO_2$_X2 | 9.59 | 5.99 | 3.84 | 3.14 | 1.99 | 1.24 |
| $NO_2$_X3 | 8.26 | 5.89 | 3.51 | 2.19 | 1.27 | 0.72 |
| $NO_2$_X4 | 11.34 | 6.25 | 3.04 | 1.17 | 0.68 | 0.33 |
| $NO_2$_X5 | 5.47 | 3.77 | 2.56 | 1.46 | 0.71 | 0.34 |
| $NO_2$_X6 | 4.70 | 2.78 | 2.19 | 0.83 | 0.48 | 0.24 |
| PAN_X0 | 9.52 | 6.41 | 4.29 | 2.98 | 1.73 | 1.04 |
| PAN_X1 | 6.42 | 3.26 | 2.59 | 1.09 | 0.66 | 0.31 |
| PAN_X2 | 8.12 | 5.76 | 3.64 | 1.23 | 0.75 | 0.48 |
| PAN_X3 | 6.22 | 4.43 | 2.96 | 0.96 | 0.60 | 0.29 |
| PAN_X4 | 5.99 | 2.96 | 1.60 | 0.75 | 0.58 | 0.37 |
| PAN_X5 | 2.08 | 1.56 | 1.01 | 0.014 | 0.0074 | 0.0023 |
| PAN_X6 | 4.97 | 2.25 | 1.33 | 0.672 | 0.41 | 0.24 |
| HONO_X0 | 4.55 | 2.21 | 1.34 | 1.52 | 1.13 | 0.93 |
| HONO_X1 | 5.49 | 2.42 | 1.87 | 1.13 | 0.79 | 0.42 |
| HONO_X2 | 4.36 | 2.78 | 1.39 | 0.65 | 0.38 | 0.21 |
| HONO_X3 | 13.89 | 9.64 | 6.05 | 1.56 | 0.81 | 0.32 |
| HONO_X4 | 8.56 | 5.14 | 3.30 | 1.61 | 0.87 | 0.48 |
| HONO_X5 | 7.41 | 4.78 | 2.39 | 1.32 | 0.95 | 0.64 |
| HONO_X6 | 5.73 | 3.67 | 1.95 | 0.41 | 0.27 | 0.071 |
| OH | 3.01 | 2.17 | 1.61 | 0.19 | 0.129 | 0.064 |
| $HO_2$ | 5.15 | 3.73 | 2.36 | 0.35 | 0.17 | 0.053 |

* Normalized error is calculated as $|C_{V2\text{-}EBI} - C_{V2\text{-}SMVGEAR}|/C_{V2\_SMVGEAR}$. This is calculated for hourly

concentrations for all the grid cells in the entire day. The mean normalized error is calculated by averaging

the normalized error for all the grid cells.

For OH and $HO_2$, the EBI solver agrees with the SMVGEAR solver very well, with

mean differences of ~0.19% and 0.35%, respectively. This indicates that the overall

gas-phase chemistry is not significantly influenced by replacing the SMVGEAR solver with

the EBI scheme. Figure 3 shows the comparison of all the hourly concentrations of OH and

$HO_2$ at hour 6, which represent average concentrations between 1400-1500 local time. The

EBI results agree with SMVGEAR solver results across all concentration ranges that span

more than three orders of magnitude. Figure 4 shows the comparison of hourly

concentrations of $NO_2$, NO, $NO_3$, HONO and PAN for hour 24 of day-1, with day-2 and

445 day-3 results presented in Figures A2 and A3 respectively. For the total concentrations these

446 species (i.e. sum of the concentrations of the source-tagged species), the source-oriented EBI

predictions agree very well with the observations. For the individual source-tagged species,

differences between the EBI and SMVGEAR results are highest among the $NO_2$, NO and

HONO species that are used to track the initial and boundary contributions (IC/BC type, first

column of Figure 4). The results shown in the plots are essentially initial concentrations

because boundary conditions for NO and $NO_2$ are quite low, and won't contribute to such

high concentrations (see Figure 6 – the regional $NO_2$ and NO plots). The source-oriented EBI

predictions for IC/BC species are biased high, while those for other tagged species are biased

low, compared to those predicted using the Gear solver. The observed discrepancies arise

because the dynamic stage solution predicted by the Eulerian backward scheme is generally

slower, exhibiting a significant time lag. In this test, the IC/BC species inherited a

concentration field from a 7-day non-source-oriented simulation, while the other tagged

species started from near-zero concentrations. Consequently, the decay of IC/BC species

(with no emission associated) was overestimated, while the accumulation of other tagged

species (associated with emissions) was underestimated. These over-predictions are balanced

by the general under-predictions of EBI for species associated with emissions, resulting in

close agreement in total concentrations. Furthermore, the errors associated with the EBI

prediction exhibit a decreasing trend with advancing flow time and diminish to within a

tolerable range (on the order of 0.1%-1%) by the end of day 3. This phenomenon suggests

that species concentrations asymptotically approach a new steady state dictated by external

inputs, with emission intensity being the primary factor.

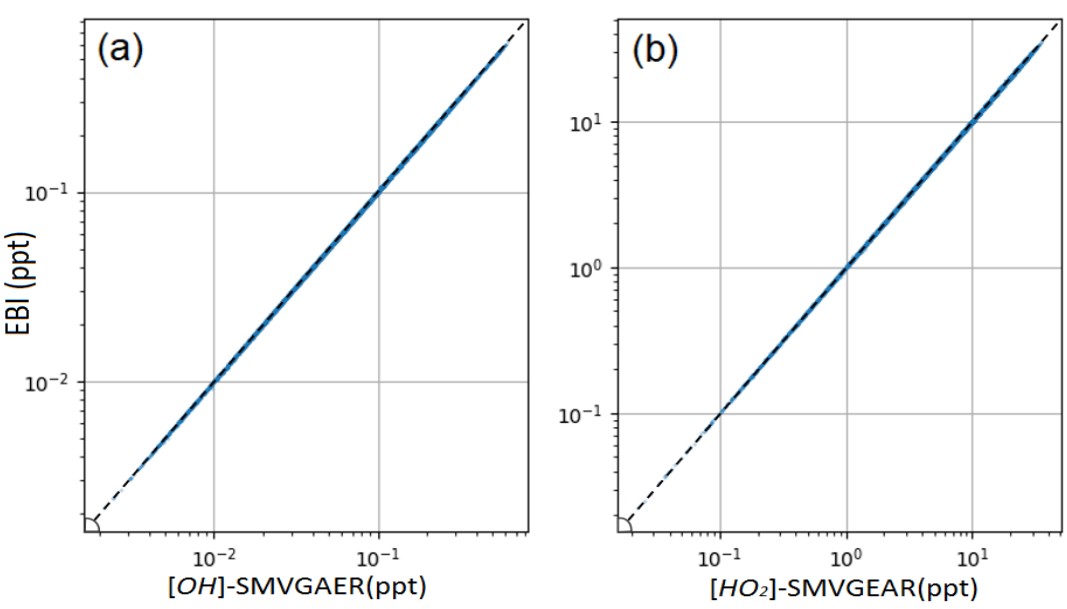

**Fig. 3.** Comparison of predicted OH (a) and $HO_2$ (b) radicals in the surface layer of the

36-km resolution domain using the source-oriented EBI(new) and the SMVGEAR(baseline)

solvers. Concentrations at hour 6 (local time 1400-1500) are shown.

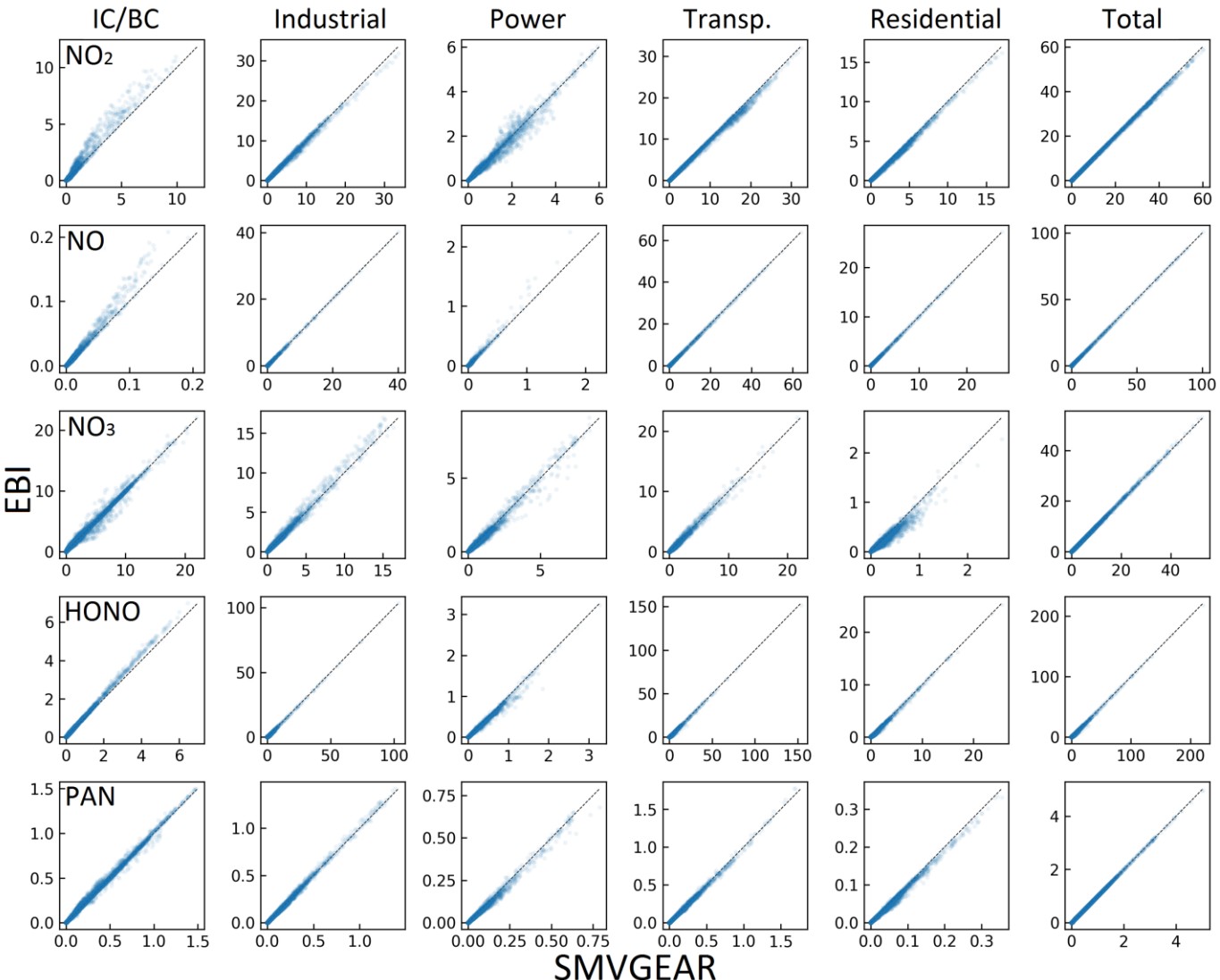

**Fig. 4.** Predicted hourly-averaged $NO_2$, NO, $NO_3$, HONO and PAN concentrations for different source types in the last hour of the day-1 simulation using source-oriented EBI(new) and the SMVGEAR(baseline) solver. Concentrations of all grid cells in the surface layer are included in the plot. Concentrations are in units of ppb for NO, $NO_2$ and PAN and in units of ppt for $NO_3$ and HONO.

Figure 5 shows the predicted hourly time series of OH, HONO, $NO_2$ and PAN at five grid cells that represent no emission, and low, medium, high and intense emission conditions, respectively. For HONO, $NO_2$ and PAN, predicted concentrations are shown for IC/BC type and the sum of the tagged species for other emission-related types. This again demonstrates that predicted OH from EBI and SMVGEAR agree with each other at all times under all emission conditions. The fraction of initial concentrations to the total concentration decreases as the emission intensity increases and the difference between EBI and SMVGEAR remains very small.

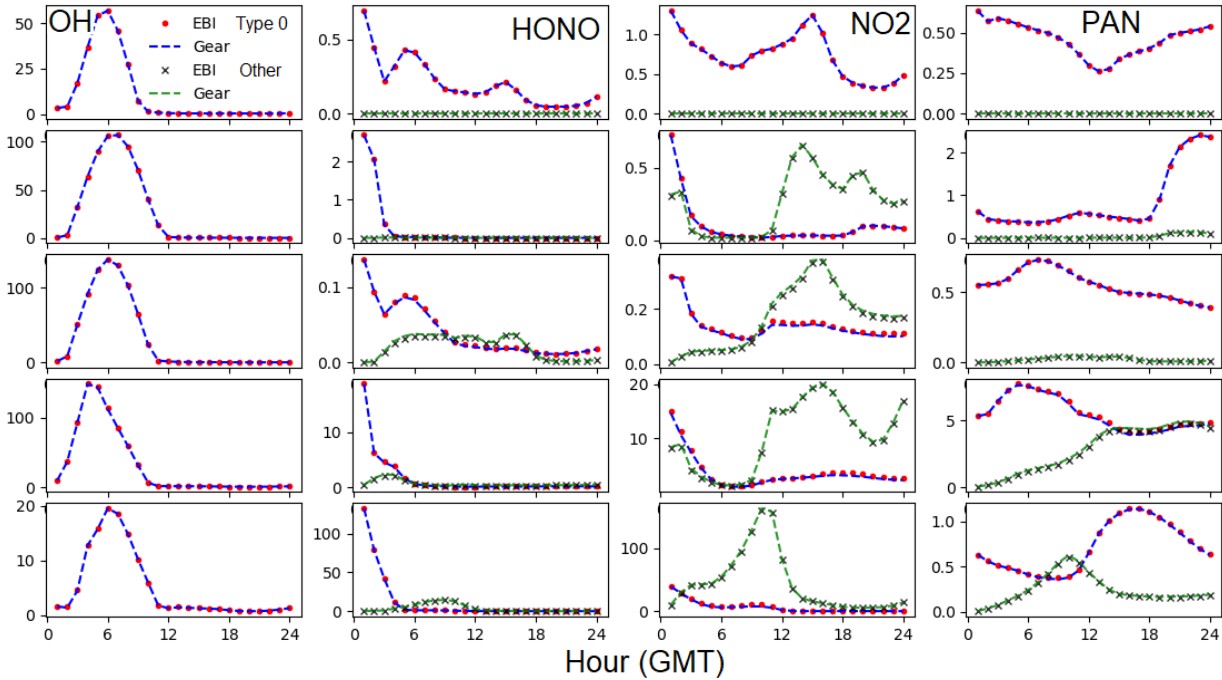

**Fig. 5.** Timeseries of OH, HONO, NO$_2$ and PAN at grid cells that represent different emission conditions (no emissions, low emission, medium emission, high emission and intense emission) for the one-day simulation. Units are ppq (parts per quadrillion) for OH, ppt for HONO and 0.1ppb for NO$_2$ and PAN. Type 0 is the concentration for the IC/BC source type and "other" represents the sum of the concentrations of all other tagged species.

Figures 6 and 7 illustrate the spatial distribution of daily average concentrations predicted from two methods. HO, HO$_2$ and two tagged NO, NO$_2$ and HONO of 36 km grid were selected as sample species, results from proposed EBI scheme are very close to the SMVGEAR results. The concentration of X0 species predicted by EBI scheme at high concentration are higher than the Gear, the reason of such slight deviation is mainly caused by the aforementioned time lag in results of Euler backward scheme.

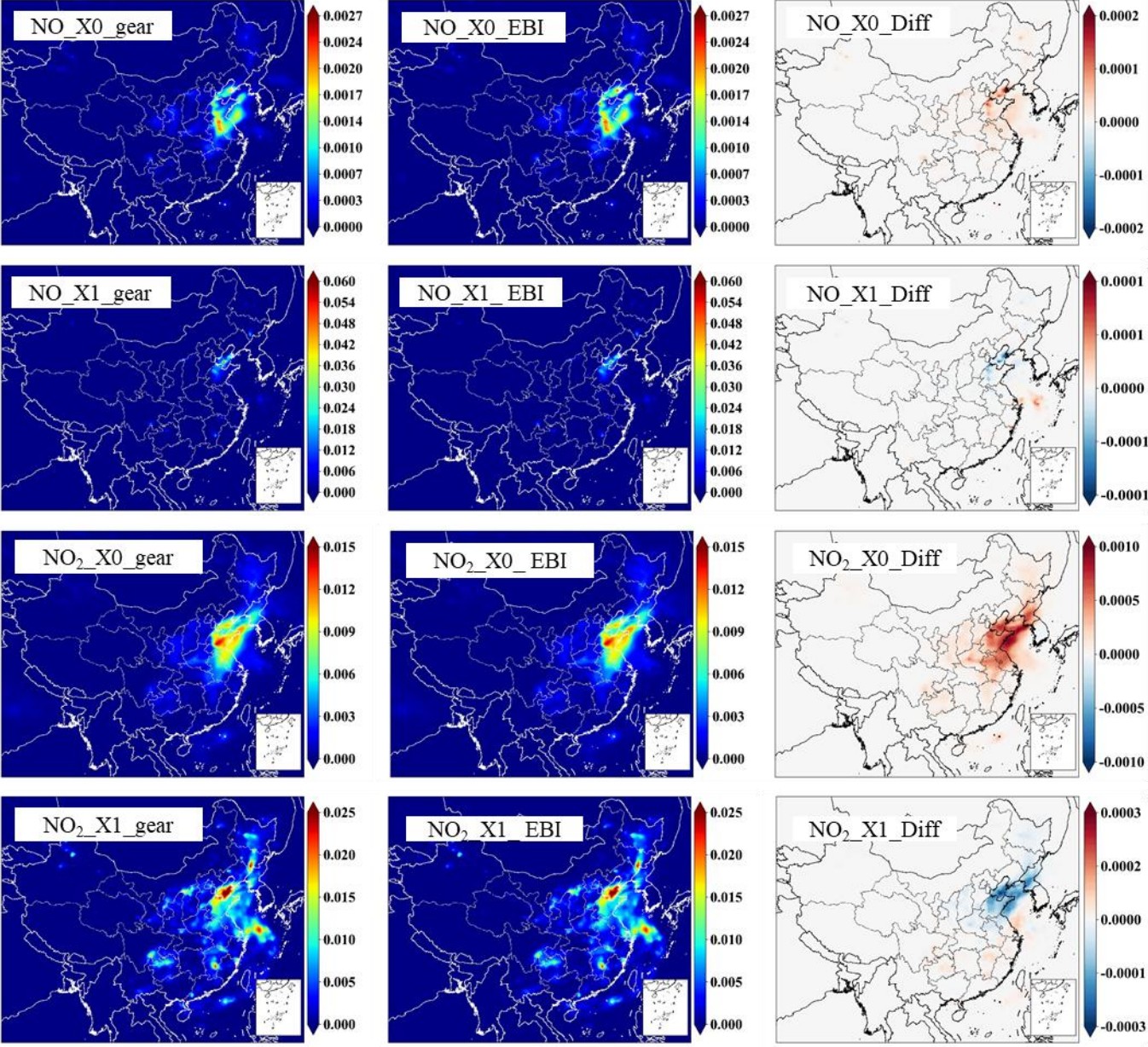

**Fig. 6.** Regional distribution of NO and NO$_2$ from a day-1 CMAQ simulation with 36km resolution and meteorological fields from WRFv4.1.4, units are ppm.

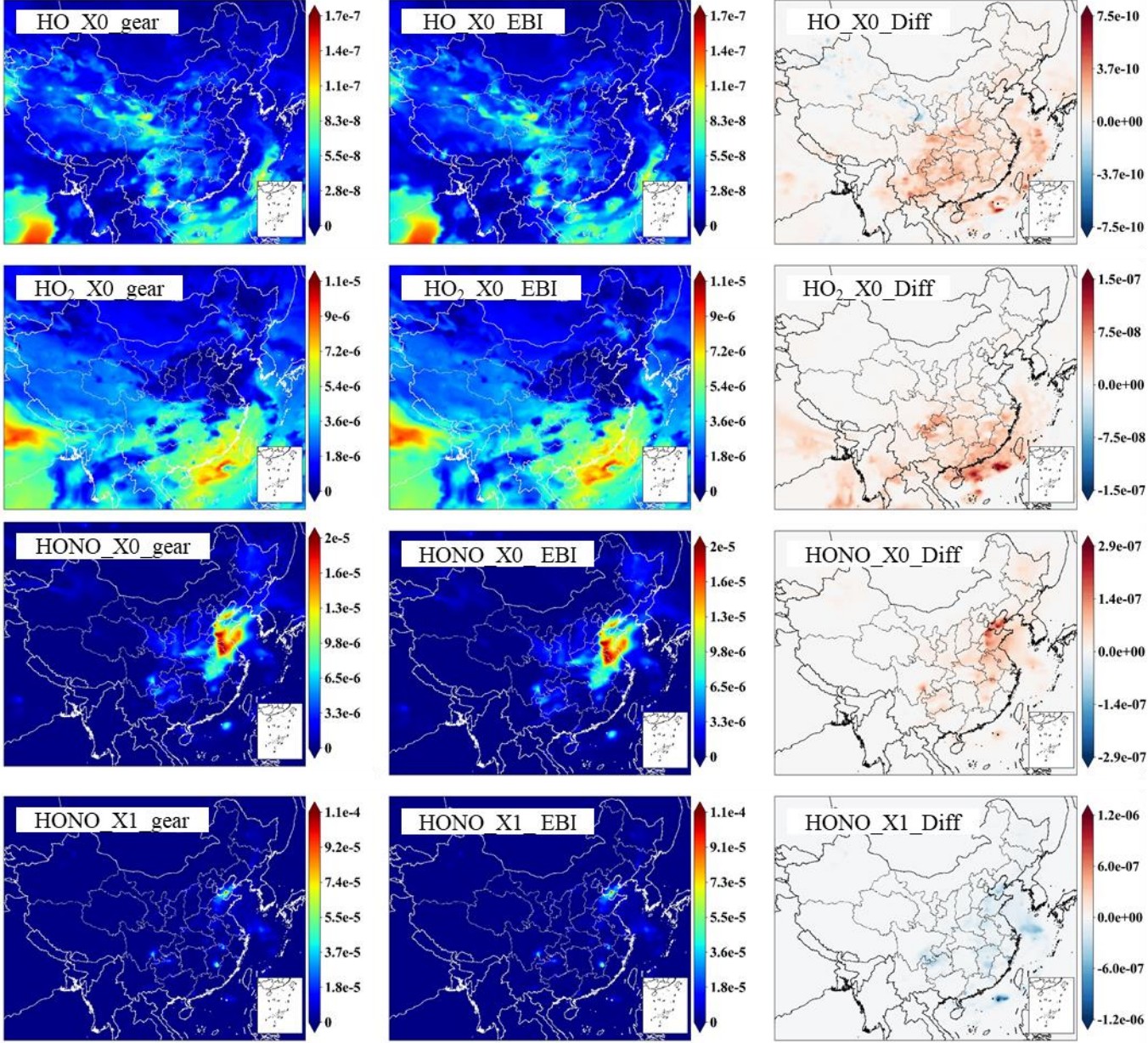

**Fig. 7.** Regional distribution of HO HO$_2$ and HONO from a day-1 CMAQ simulation with
160X130 36km resolution domain and meteorological fields from WRFv4.1.4, units are
504 ppm.

**4 Conclusions**

In this study, the computation efficiency and thus scalability of the source-oriented
approach is greatly improved with a new approach of dealing with these two-tagged-species
reactions. The new approach is based on tracking the total concentration of the source-tagged
species and reduce the n$^2$ number of second-order reactions into 2n pseudo first-order
reactions for chemical system with n sources, this method preserves individual species'
production and loss rates, thus leading to improved computational efficiency because the
total number of reactions increases linearly with the source number. Additionally, Euler
Backward Iterative (EBI) solver has been successfully implemented to the source-oriented
mechanism, with average of absolute relative error is below 5% and up to 90% chemistry

time reduction in comparison to SMVGEAR. While efficient source-oriented approach for primary particles are already available to track a large number of sources simultaneously, the efficient approach developed in this study has the potential to track a large number of sources to evaluate their impact on secondary pollutant formation, and has the potential to be applied in air quality forecasting models that provide source or source-region contribution information for policy makers for better emission regulations under meteorological conditions that exacerbate pollution.

**Data availability.** Data can be obtained upon request from the authors.

**Authorship contributions.** QX: Investigation, Methodology, Software, Visualization, Writing & editing. ZJ: Visualization, Writing - review & editing. QY: Software, Data curation, Validation, Visualization, Writing - review & editing. KW: Supervision, Writing - review & editing, Visualization. FS: Writing, Investigation, Conceptualization, Methodology, Software, Validation. RZ: Resources, Supervision, Visualization. M. K: Software, Methodology.

**Declaration of competing interest.** The contact author has declared that neither they nor their co-authors have any competing interests.

**Financial support.** This work was supported by National Key Research and Development Program of China (No. 2024YFC3713703) and Study of Zhengzhou $PM_{2.5}$ and $O_3$ Collaborative Control and Monitoring Project (No. 20220347A).

## Acknowledgments

The development of the simplified representation of the source-oriented reactions is originally funded by the Texas Air Quality Research Program (AQRP) (10-010). The authors would like to thank Texas A&M High Performance Computing Center for providing the computation resources.

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

## Appendix

**List A1.** The special reaction rate section (between the keywords SPECIAL and END SPECIAL) and reactions used to implement the source-oriented $NO_2+NO_3$ reactions in dual-tagged reaction reduction, with 10 source types using the chemical mechanism preprocessor CHEMMECH for the CMAQ model. Due to the limitation of the current mechanism preprocessor, a dummy reaction (<10_dum>) is needed so that the original reaction rate can be included in calculation of the special rate constants. Using the special rate expression is signaled by including the symbol '?' in the reaction rate coefficient expression.

```
SPECIAL =
RNO_NO3 = K<10_dum>*C<NO3> + K<10_dum>*C<NO3_X1> +
<10_dum>*C<NO3_X2>
      + K<10_dum>*C<NO3_X3> + K<10_dum>*C<NO3_X4> +
K<10_dum>*C<NO3_X5>
      + K<10_dum>*C<NO3_X6> + K<10_dum>*C<NO3_X7> +
K<10_dum>*C<NO3_X8>
      + K<10_dum>*C<NO3_X9>;
RNO3_NO = K<10_dum>*C<NO> + K<10_dum>*C<NO_X1> +
K<10_dum>*C<NO_X2> +
        K<10_dum>*C<NO_X3> + K<10_dum>*C<NO_X4> +
K<10_dum>*C<NO_X5> +
        K<10_dum>*C<NO_X6> + K<10_dum>*C<NO_X7> +
K<10_dum>*C<NO_X8> +
        K<10_dum>*C<NO_X9>;
...
END SPECIAL
...
<10_dum> dummy1 + dummy1 = dummy1 + dummy1 #1.80e-11@-110;
<10_aX0> NO = NO2 #1.0?RNO_NO3;
<10_aX1> NO_X1 = NO2_X1 #1.0?RNO_NO3;
<10_aX2> NO_X2 = NO2_X2 #1.0?RNO_NO3;
...
<10_aX9> NO_X9 = NO2_X9 #1.0?RNO_NO3;
<10_bX0> NO3 = NO2 #1.0?RNO3_NO;
<10_bX1> NO3_X1 = NO2_X1 #1.0?RNO3_NO;
<10_bX2> NO3_X2 = NO2_X2 #1.0?RNO3_NO;
...
<10_bX9> NO3_X9 = NO2_X9 #1.0?RNO3_NO;
```

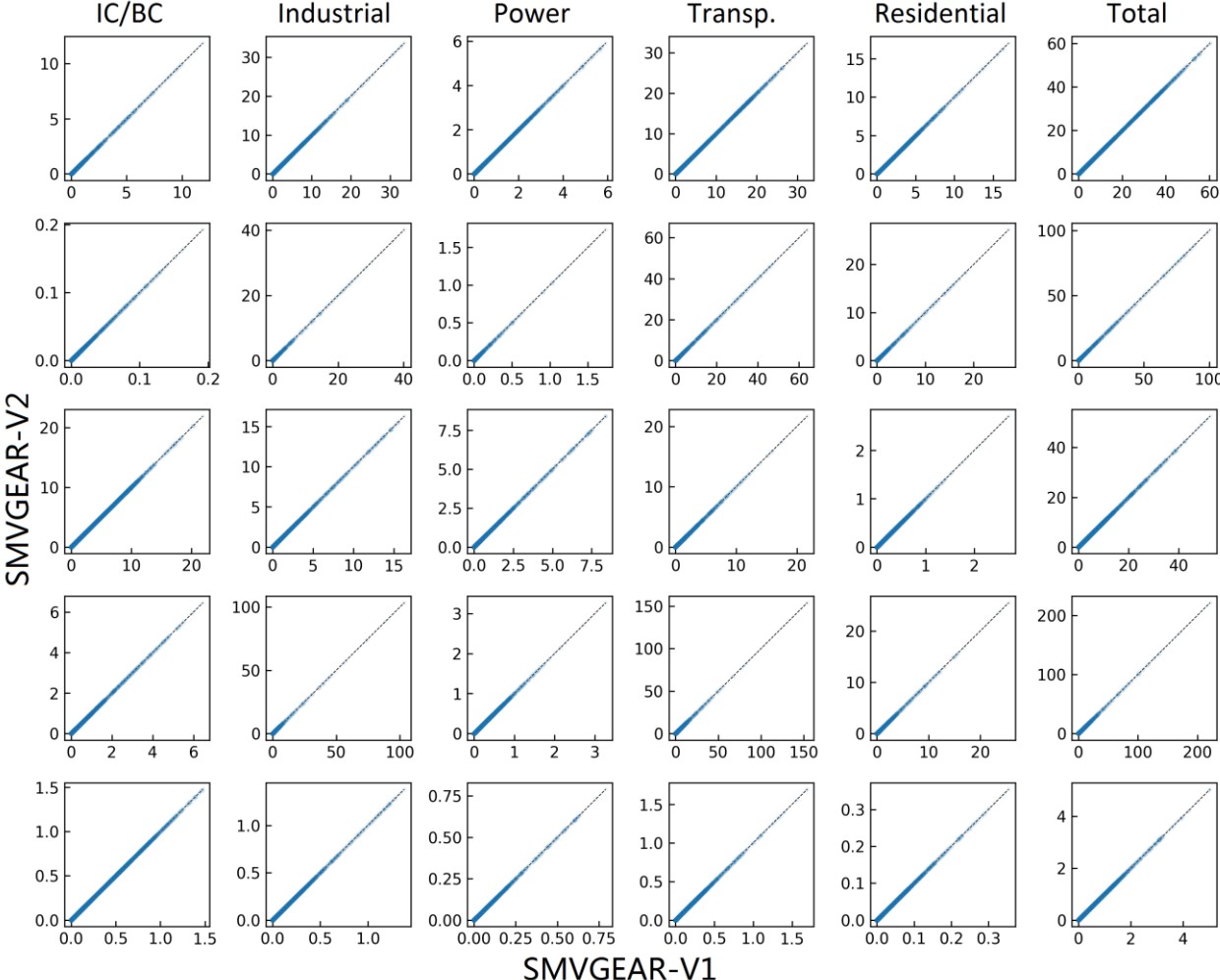

**Fig. A1.** Predicted hourly-averaged $NO_2$, NO, $NO_3$, HONO and PAN concentrations for different source types in the last hour of the day-1 simulation using SMVGEAR solver with fully expanded source-oriented (V1) and the dual-tagged reaction reduction method (V2) mechanism. Concentrations of all grid cells in the surface layer of 160X130 36km resolution domain are included in the plot. Concentrations are in units of ppb for NO, $NO_2$ and PAN and in units of ppt for $NO_3$ and HONO.

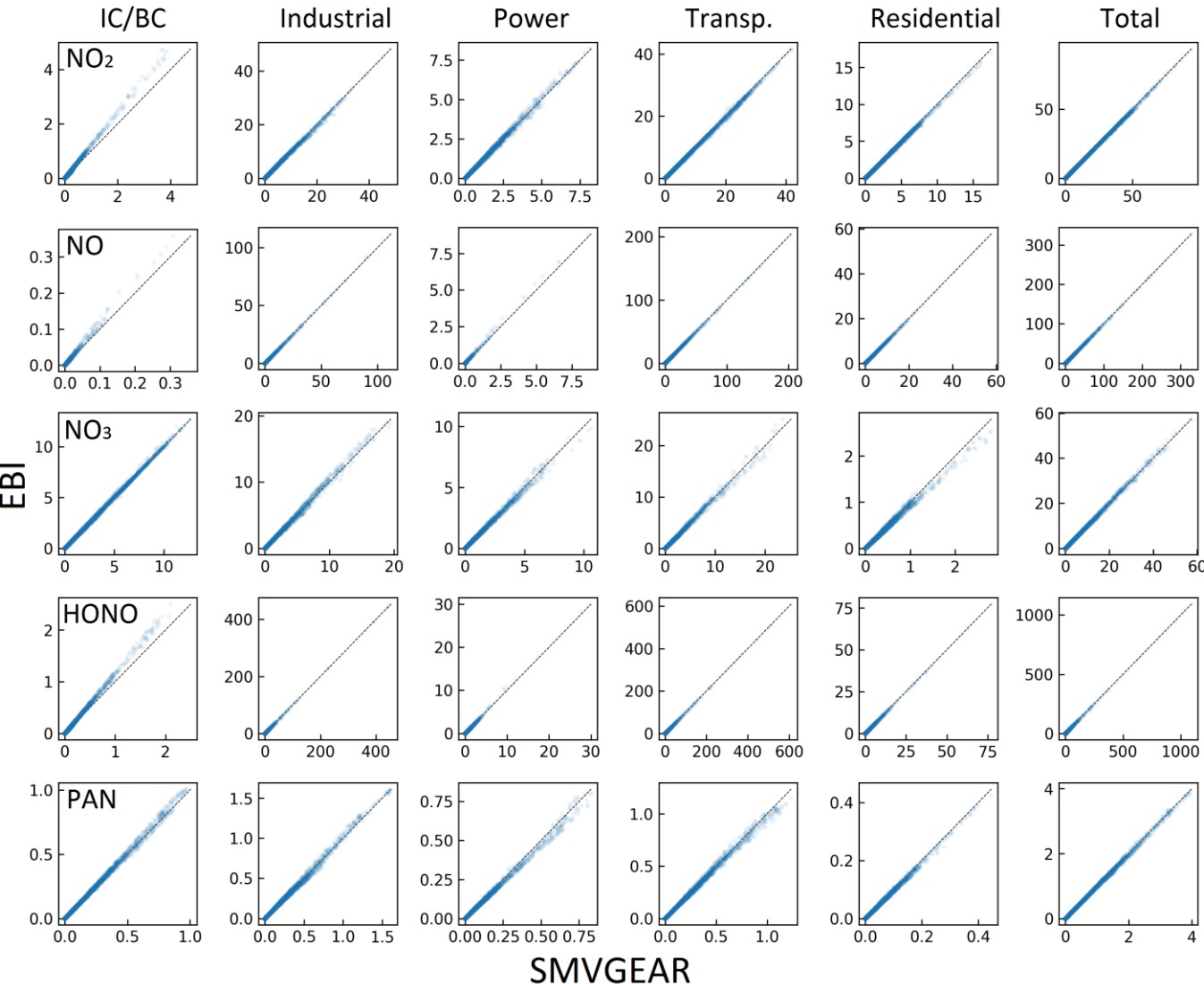

**Fig. A2.** Predicted hourly-averaged $NO_2$, NO, $NO_3$, HONO and PAN concentrations for different source types in the last hour of the day-2 simulation using source-oriented EBI(new) and the SMVGEAR(baseline) solver. Concentrations of all grid cells in the surface layer of 160X130 36km resolution domain are included in the plot. Concentrations are in units of ppb for NO, $NO_2$ and PAN and in units of ppt for $NO_3$ and HONO.

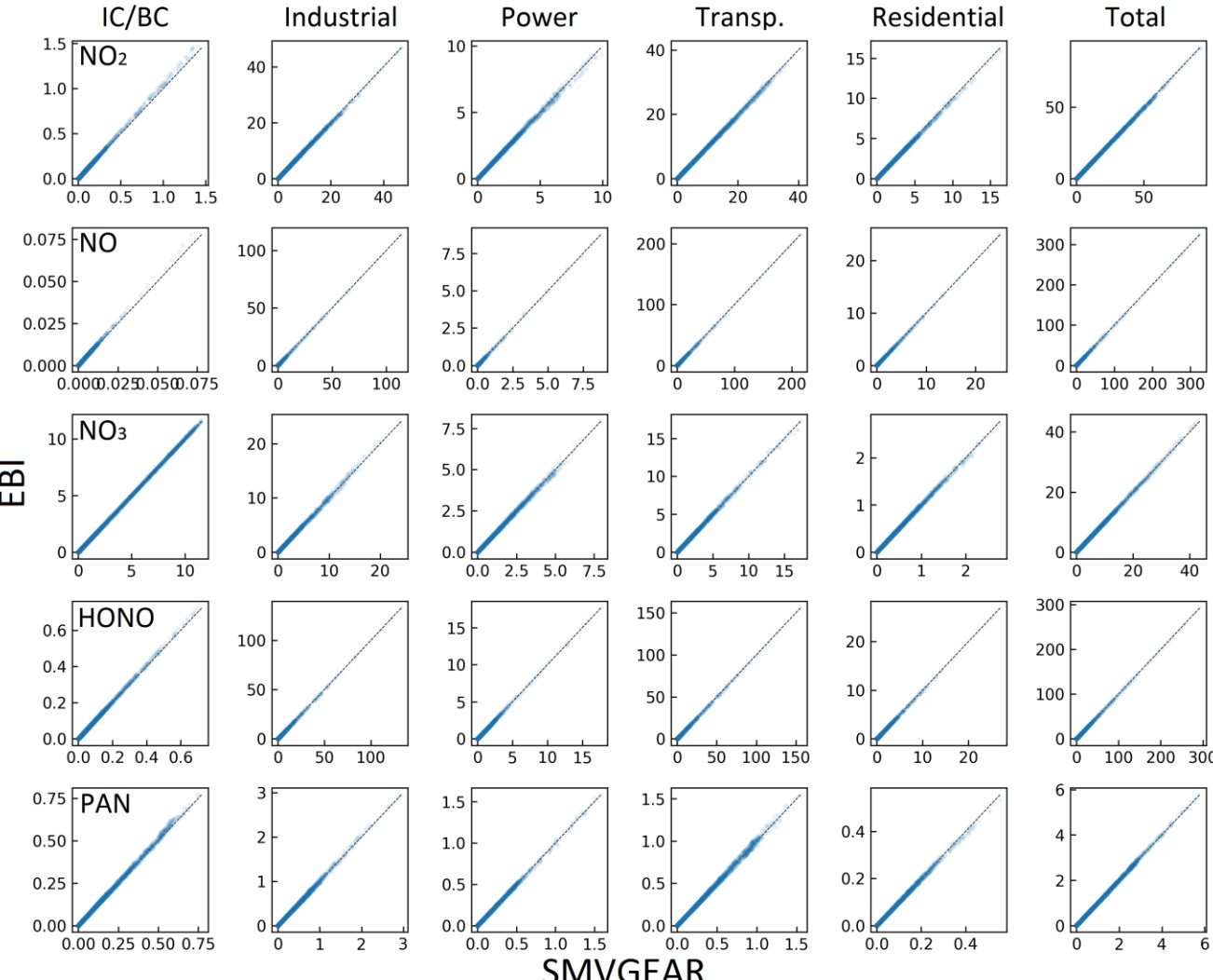

Fig. A3. Predicted hourly-averaged $NO_2$, NO, $NO_3$, HONO and PAN concentrations for different source types in the last hour of the day-3 simulation using source-oriented EBI(new) and the SMVGEAR(baseline) solver. Concentrations of all grid cells in the surface layer of 160X130 36km resolution domain are included in the plot. Concentrations are in units of ppb for NO, $NO_2$ and PAN and in units of ppt for $NO_3$ and HONO.