# Peer review of "Improving the computation efficiency of a source-oriented chemical mechanism for the"

_EGUsphere, 2025_

## Author Comment (AC1)

**Manuscript number:** EGUSPHERE-2025-44

**Title: Improving the computation efficiency of a source-oriented chemical mechanism for the simultaneous source apportionment of ozone and secondary particulate pollutants**

**Reviewer comment #1:**

Xu et al. develop a source-oriented chemical mechanism that with tags that is computationally efficient. A novel and useful contribution is the ability to retain the source tags through secondary chemistry with complexity that scales linearly with number of sources, rather than quadratically.

An algorithm that reduces the computational complexity from quadratic to linear with number is in itself a useful contribution (analogous complexity-linearized algorithm have been very useful in other subfields of atmospheric chemistry and physics, e.g. the superdroplet method). This approach is also well explained in section 2.1. However, the authors dedicate much of the paper to the Euler Backward Iterative (EBI) solver, where the motivation, methodology, and novelty is still unclear after several readthroughs. In a revision, the authors should make clear why EBI can't be applied to the fully expanded mechanism in V1, and make clearer what its contribution is and how the EBI relates to the linear reduction method.

I believe this paper could be a good fit for ACP (or perhaps the related journal GMD), but should address a few major issues that can be resolved with clarifying a few sections.

Below are specific questions and suggestions, divided into major and minor comments.

**Response:** We would like to express our sincere gratitude for the detailed and constructive feedback on our manuscript from the eviewer's positive comments and valuable suggestions. We confirm that all your concerns and suggestions have been fully addressed in the revised manuscript. This includes modifications to the main

technical issues, enhancements to the figures and tables, as well as substantial improvements to the clarity and precision of the language throughout the text. Specifically, revisions include clarifications regarding the main purpose of this study, EBI scheme's application and limitations, enhanced explanations of the proposed methods (dual-tagged reaction reduction and successive under-relaxation), corrections to equations and terminology, more detailed analysis of the dynamic under-relaxation factor, and additional discussion of error propagation over time. Below are the point-to-point responses to all of thecomments (The comments are marked in black color and the responses are markedin dark blue color). Note that the following line numbers are shown in the corrected version.

**Major Comments:**

Comment 1: Line 22: There are a few sentences here that are copied and pasted verbatim into the conclusions. While these are the key points, I would discourage exact repeating of text in different parts of the manuscript.

Response: Thanks for the comment. The duplicated sentence has been modified to: "The new approach is based on tracking the total concentration of the source-tagged species and reduce the $n^2$ number of second-order reactions into 2n pseudo first-order reactions for chemical system with n sources, this method preserves individual species' production and loss rates, thus leading to improved computational efficiency because the total number of reactions increases linearly with the source number." (Line 508–512) in the conclusion.

Comment 2: Lines 130-137: In this paragraph, it is unclear on how the EBI solver is incompatible with source-oriented mechanisms (without the linear reduction approach introduced in this paper). This is important motivation for the rest of the paper, but is not clear. What is a "non-typed regular mechanism" ? What does "applicability" mean, specifically? By "explicit solutions", do the authors mean closed-form solutions, e.g. algebraic solutions derived from pseudo steady-state assumptions, or explicit Euler solvers, e.g. forward Euler solvers with many small explicit timesteps? Refining the

language in a revision may help with clarity, and clearly explaining the limitations of applying EBI to source-tagged mechanisms is important for motivating this study.

Response: Thanks for the comment and sorry for the misunderstanding. The original version of the EBI scheme was designed to calculate the total concentrations of species, with reactive nitrate species strongly involve in the photochemical reactions such as NO, $NO_2$, $O_3$, HONO, their total concentration calculated as analytical solutions, by several steps including a set of complicated procedures for the non-linear algebraic equations (Hertel et al., 1993), rather than the generalized Eulerian backward scheme (equation (2)). In comparison to SMVGEAR solver, the virtue of EBI solver is speeding, as the computational cost is way lower than the former. However, the original EBI scheme could not be directly used for source-tagged mechanisms, because the analytical solution procedures for photochemical species in the EBI scheme are sophisticated, hard-coded and for total concentrations of species only, which inherently excludes source-tagged species. The words "non-typed regular mechanism" means the original mechanism, in which all species are referring to the total species regardless the source tagged, for instance the specie with name $NO_2$ indicates all $NO_2$ in the original mechanism. Contrarily, in the source-oriented mechanism, the $NO_2$ refers to only the $NO_2$ with null source tag, while the $NO_2\_Xi$ indicates the $NO_2$ relates to the $i^{th}$ source tag. Therefore, the original EBI scheme will be incompatible to the source-oriented mechanism, as the species other than null source tagged species will be excluded. Direct application of the original EBI scheme to the source-tagged mechanism results in incorrect solutions for null source tag species, even though they are included, because the underestimation of reaction rates neglects the contribution of source-tagged reactions. The "applicability" means with the capability of handling the source-oriented mechanism, the EBI solver could be used for tasks which requires timing efficiency such as air quality forecast scenario with source-oriented mechanism, if the concentrations of species related with certain emission sector are interested. Juxtaposing the words 'efficiency' and 'applicability' is not quite appropriate, as the latter is often a consequence of the former. The term "explicit solutions" refers to the

analytical solutions of total concentration of reactive nitrate species within the four special groups. These solutions are calculated directly using four sets of non-linear algebraic equations (e.g. Eq. 4a-4d in manuscript, also mentioned in Line 216). These analytical solutions, unlike implicit or iterative methods, employs a sophisticated solution procedure involving bunch of mathematical manipulation, as demonstrated in the work of Hertel et al. Revision has been made upon the mentioned points in the second last paragraph of section 1.2. (Line 140–152)

Comment 3: Equations 3a and 3b: This approach only works for bimolecular reaction rates, is that correct? It seems like the separation by source in this way is only compatible with the mass action law because of the bilinearity of k[Atot][Btot]. This would be good to acknowledge, or if compatible with other rate laws or reaction orders, perhaps include a note of how it is compatible.

Response: Thanks for the comment and sorry for the misunderstanding. This approach targets bimolecular reaction rates involving double source tagged reactants, which constitute a minority of the reactions within this source-tagged mechanism. Single tagged reactions, on the other hand, are the majority of source-tagged reactions. The reactants separation method remains valid as long as the reaction rate is proportional to the concentration or a power of the concentration of individual species (A) or a combination of species (A and B). Given the extremely low concentrations of atmospheric species, apparent mechanisms for atmospheric chemistry are typically linear or bi-linear, ensuring the compatibility of this method in current and future mechanisms. Consequently, the separation method is applicable because terms other than [A] can be lumped into the effective reaction rate constant, a common practice in atmospheric chemistry studies. Extra description added "This method is valid for reactions with rate laws of the form rate=$k[A]^m[B]^n$..., where the rate is proportional to the product of reactant concentrations raised to their orders." to illustrate the compatibility of separation method to other forms of reaction rate expressions. (Line 174–176)

Comment 4: Lines 253-256: I don't understand this sentence. Is there a separate solver to predict the total concentrations, and then the ratios between the summed tagged concentrations and independently solved total concentrations used to scale each tagged concentration?

Response: Thanks for the comment and sorry for the misunderstanding. Yes, this might cause some ambiguity. Total concentrations are not calculated by a separate solver; rather, the original EBI calculates them prior to solving for source-tagged species. Photochemical species (e.g., NO, $NO_2$, $O_3$, $O_3P$) are calculated using the analytical solution subroutines mentioned earlier. Once these total concentrations are determined (as in 4a-4d), the source-tagged species are then solved (e.g., in 5a & 5b). For instance, solving for source-tagged NO and $NO_2$ requires the concentration of $O_3$, which appears on the right-hand side of 5a & 5b. Notably, the total concentrations of NO, $NO_2$, $O_3$, and $O_3P$ (from 4a-4d) are determined by the aforementioned explicit method. Therefore, these total concentrations are a prerequisite for predicting source-tagged NO and $NO_2$. The purpose of this readjustment is to avoid numerical errors arising from the inversion of ill-conditioned matrices (6a-6c), a common issue in source apportionment calculations due to significant differences in the magnitudes of tagged species concentrations. Additional description has been added in the introduction "In the EBI solver, four family groups of strongly coupled species, are excluded from the general equation (2): (1) NO, $NO_2$, $O_3$ and $O(^3P)$, (2) OH, $HO_2$, HONO and $HNO_4$, (3) peroxyacetyl radical (CH3C(O)OO·, or $C_2O_3$) and peroxyacetyl nitrate (PAN), and (4) $NO_3$ and $N_2O_5$. For these species, analytical solutions of four sets of non-linear algebraic equations (Eqs. 9-12 in Hertel et al., 1993) are applied to determine their concentrations at time t instead of using the P and L terms with equation (2). The detailed mathematical procedures involved in obtaining these analytical solutions are listed in Appendices A.1-A.4 of Hertel et al. (1993)." (Line 130–137)

Comment 5: Line 260: The use of "final" here is ambiguous. Would it be more accurate to say "current iteration"? The use of t in the subscript already implies final

in terms of time (though the authors refer to this as current timestep, which also seems appropriate). But from what I understood from the results in 3.1, this must be applied at every iteration up to convergence, not just an adjustment applied post convergence to the solution to make the "final" solution ~20% (or 1-alpha) more like the previous timestep? This would be important to make clearer in a revision.

Response: Thanks for the comment and sorry for the misunderstanding. Yes, this might cause ambiguity. Regarding the term "final," which refers to the concentration value being updated in the solution at the end of an iteration, it should be changed to "updated." (Line 281) The modification using values from the previous iteration is introduced to avoid the non-converging oscillations characteristic of fixed-point iteration. Furthermore, the description in equation 11 are inaccurate. The under-relaxation method updates the solution vector at the end of an iteration based on values from the previous iteration and the predicted values in the current iteration, not values from old and current time steps. Consequently, all instances of "time step" should be replaced with "iteration," including in Eq. (11). And the word "final" changed to "updated", and corrections are made in line 279-285 as well as in Eq. (11).

Comment 6: Fig 2 and table 2: This is a very interesting result, and a revision could go in more depth on these results, which would be a major addition to the study. I noticed the dynamic under-relaxation factor is not fully monotonic, is there a reason for that? More generally, how did the authors obtain the values for alpha shown in table 2? Is there any explanation why the solution in later *iterations* should weight the previous *timestep* more heavily? Finally, do the authors have any explanation (or intuition) for why a dynamic under-relaxation factor converges faster than a constant alpha?

The dynamic under-relaxation factor is not fully monotonic, is there a reason for that? More generally, how did the authors obtain the values for alpha shown in table 2? Is there any explanation why the solution in later iterations should weight the previous timestep more heavily? Finally, do the authors have any explanation (or intuition) for why a dynamic under-relaxation factor converges faster than a constant alpha?

Response: Thanks for the valuable comment. Yes, an in-depth discussion should be added. The under-relaxation approach is commonly employed in commercial Computational Fluid Dynamics (CFD) codes to mitigate numerical instability in highly non-linear systems. For under-relaxation of scalar variables such as concentration or energy, a recommended $\alpha$ value is 0.8, consistent with the optimal fixed $\alpha$ shown in Fig. 2. However, a dynamic $\alpha$ generally offers better performance due to its greater flexibility. Analysis of predicted solutions without under-relaxation reveals that in the initial iterations, the solution vector rapidly shifts towards the vicinity of the true solution before oscillating around it until convergence is achieved. Therefore, to accelerate convergence, an aggressive selection of $\alpha$ (larger value) is beneficial in the early iterations to quickly propel the solution vector towards this near-optimal space. In later iterations, a more conservative selection of $\alpha$ (smaller value, weighting the previous iteration more heavily) is preferred to update the solution vector slowly towards the true solution, gradually eliminating residual errors and effectively damping out slight overshoots. This initial "kick" provided by the larger initial $\alpha$ explains why a dynamic $\alpha$ often converges faster than a fixed $\alpha$, which lacks this initial boost. This behavior is analogous to a damped oscillatory system, where critical or slightly under-damped systems achieve equilibrium faster than over-damped or highly under-damped ones, which are either too slow or prone to instability.

The $\alpha$ values in Table 2 were derived from the aforementioned concepts and fine-tuned through several numerical experiments. The non-monotonic behavior of $\alpha$ is attributed to the fact that for most test cases, convergence was achieved within 10 iterations. In the less frequent cases which require more iterations, a slightly larger adjustment step could accelerate the approach to the true solution. The $\alpha$ value of 0.79 for iterations greater than 16 represents a compromise between rapid movement and effective damping of overshoots. And in-depth discussion has been added to the relevant places. (Line 398–409)

**Minor Comments:**

1. Line 25 and the copied phrase in the conclusion: improved -> improving

   Response: Thanks for the comment, "improved" has been changed to "improving". (Line 24)

2. RS1: To distinguish source from stoich, I would suggest using superscripts for tags or operators $(NO)\_S1 + (NO\_3)\_S1$

   Response: Thank you for your suggestion. Subscript for source tag in RS1 changed to superscript. (Line 51)

3. RS2: Would a different example with tagged products be better here, to illustrate the relevance? I know this is just an example, but why have different reactions if RO2_R and RCHO products are the same species for every reaction? This approach is more useful than just for estimating different sinks of reactants.

   Response: Thank you for your suggestion. A different example with $TERP + NO_3$ has replaced the previous one. (Line 70)

4. Line 59: Is this "also" as in additionally, or is it specifically for the reason in the previous sentence? If also refers to species quadratic scaling as well as reactions, maybe this point could be made right before the N2O5 example.

   Response: Thanks for the comment. Also means increase in both number of reactions and tagged species, a clarification has been made before "also" in the manuscript. The sentence in manuscript has been modified to: "The necessity to deal with $N_2O_5$, which can be generated from $NO_2$ and $NO_3$ from different sources, is handled with double-source-tagged species $N_2O_{5,Sij}$. In addition to a potential quadratic scaling of the number of reactions, the number of $N_2O_5$ species also increases quadratically with the number of explicit sources, leading to near-quadratic growth of the overall number of species when the number of types to track gets higher." (Line 56-61)

5. Line 94: Jacobin -> Jacobian

   Response: Thanks for the comment and sorry for the mistake. Jacobin has been changed to Jacobian. (Line 94)

6. Line 121: I would suggest reminding the reader here of $\Delta C\_t^{\wedge}(m+1)$ here, to avoid ambiguity between changes in C over time versus changes in C over iteration.

Response: Thank you for your suggestion. C has been changed to $C_{i,t}$ for a new iteration. (Line 126)

7. Line 127: The use of explicit here is confusing (see the major comment). The authors can disambiguate in many ways, e.g. if explicit solutions mean closed-form algebraic expressions arising from pseudo steady-state assumptions, say that. In the context of iterative solvers, explicit often refers to forward Euler methods versus implicit backward Euler methods.

Response: Thanks for the comment, "explicit" has been changed to "analytical" (Line 134)

8. Line 212: unknows -> unknowns

Response: Thanks for the comment and sorry for the mistake. The "unknows" has been changed to "unknowns". (Line 231)

9. Line 237: quadradic -> quadratic

Response: Thanks for the comment and sorry for the mistake. The "quadradic" has been changed to "quadratic". (Line 257)

10. Table 1: Maybe some point in the text explain V2-S is V2 with SMVGEAR?

Response: Thanks for the comment. V2-S has been explained with additional words "# V2-S: V2 reaction mechanism with SMVGEAR solver" underneath the Table 1. (Line 333)

11. Line 324-325: "For most of the species, a relative tolerance of 1e-3 is used". What are the exceptions, and why do they differ?

Response: Thanks for the comment. The convergence criteria are 1e-3 for most of species that are in differential forms, however some pseudo steady state species such as $O_3P$ and $O_1D$, the convergence criteria are 1.0, indicate convergence check not applicable. Explanation has been added "For most of the species, a relative tolerance of $1 \times 10^{-3}$ is used. Exceptions include pseudo-steady-state species like $O_3P$ and $O_1D$, for which a tolerance of 1.0 is applied, indicating that a convergence check is not applicable." (Line 353–355)

12. Line 359: "for more than three folds" was confusing at first. Not sure if it refers to the 74% figure comparing total time to V2-S, or some other comparison in Table 1.

Maybe this could be a little clearer where this conclusion is coming from. I might also suggest wording such as "by more than threefold" or "by a factor of 3 or more".

Response: Thanks for the comment, "for more than three folds" has been changed to "by a factor of 3 or more". (Line 390)

13. Line 402: "than" could be substituted with "compared to" or something similar.

Response: Thanks for the comment. The word "than" has been substituted with "compared to". Also, this part has been rewritten for in-depth discussion of the errors. (Line 454)

14. Line 432: This sentence might work better without "zone"

Response: Thanks for the comment. We had deleted the word "zone". (Line 494)

15. Line 450: "needs only 11 times of the computation time" Is this the first timing comparison to the non-source-oriented mechanism in the manuscript? This result would be good to report before the conclusions, though fine to reiterate it in the conclusion. For a frame of reference, it would be good to compare the other simulations to the untagged mechanism computation time, at least V1, and ideally V2-S as well. If including this result, why not include it in Table 1?

Response: Thanks for the comment. The phrase, "Test cases based on the Texas Air Quality Study 2006," was mistakenly placed in the conclusion and has since been moved to the introduction, where it is now properly referenced. Additional data regarding the timing of untagged mechanism with SMVGEAR solver have been added into Table 1. (Line 328-333)

---

## Author Comment (AC2)

**Manuscript number:** EGUSPHERE-2025-44

**Title: Improving the computation efficiency of a source-oriented chemical mechanism for the simultaneous source apportionment of ozone and secondary particulate pollutants**

**Reviewer comment #2:**

Overall, this is a strong paper which demonstrates a new method that has the potential to be useful in the field of study. In this paper you have two key achievements. The development of a method to reduce the number of reactions needed for source apportionment, and the implementation of an EBI solver for source apportionment. One difficulty with reading through the paper was identifying which method was used. I suggest giving a name to your method of reaction reduction, such as 'relative rate reaction reduction' (or something shorter), to be able to specify when you have used this method. Otherwise, it is often unclear that both methods have been used.

In addition, it is my understanding that the reaction reduction method does not actually add any error, yet you only show error when including the EBI. Given that this reaction reduction is a new method, I would suggest including some result showing that there is no error from reducing the number of reactions as you have done when the original SMVGEAR solver is used with the new reactions.

Unless my understanding is incorrect, these two methods do not need to be used in tandem, and it seems that the reaction reduction could be implemented relatively easily without major modification to a run, so I suggest highlighting that these methods can be used separately but are complementary of each other.

In your discussion of method error, you show that EBI has a tolerable level of error on average across the area measured. However, I think that more detail would be helpful. The biggest concern when introducing a new solver is the propagation of error over time. Here, your runs only go for 24 hours. It would be helpful to justify

why you chose that run length. Furthermore, it would be helpful to include some discussion of how the error changes over time. Does the error increase as the run goes on, or is it relatively stable?

**Response:** We would like to express our sincere gratitude for the detailed and constructive feedback on our manuscript from the eviewer's positive comments and valuable suggestions. We confirm that all your concerns and suggestions have been fully addressed in the revised manuscript. This includes modifications to the main technical issues, enhancements to the figures and tables, as well as substantial improvements to the clarity and precision of the language throughout the text. Specifically, revisions include clarifications regarding the main purpose of this study, EBI scheme's application and limitations, enhanced explanations of the proposed methods (dual-tagged reaction reduction and successive under-relaxation), corrections to equations and terminology, more detailed analysis of the dynamic under-relaxation factor, and additional discussion of error propagation over time. Below are the point-to-point responses to all of thecomments (The comments are marked in black color and the responses are markedin dark blue color). Note that the following line numbers are shown in the corrected version.

**General comments:**

Comment 1: I suggest giving a name to your method of reaction reduction, such as 'relative rate reaction reduction' (or something shorter), to be able to specify when you have used this method. Otherwise, it is often unclear that both methods have been used.

Response: Thank you for your suggestion. The name should be 'dual-tagged reaction reduction', and the name has been added in proper places in the text.

Comment 2: In addition, it is my understanding that the reaction reduction method does not actually add any error, yet you only show error when including the EBI. Given that this reaction reduction is a new method, I would suggest including some result showing that there is no error from reducing the number of reactions as you

have done when the original SMVGEAR solver is used with the new reactions.

Response: Thanks for the comment. The result comparison of the dual-tagged reaction reduction method is being added as the appendix Figure 1 and highlighted in the manuscript at the end of paragraph 2 of 2.3. (Line 323-326)

Comment 3: so I suggest highlighting that these methods can be used separately but are complementary of each other. (Reaction reduction and implement of source apportionment on EBI, highlighting that these two methods can be used separately but are complementary of each other.)

Response: Thanks for the comment. Yes, these two methods are not necessary to be used simultaneously, however they both contribute to computational cost reduction. The mechanism file generated by the CHEMMECH module from tagged-mechanism reduction method could be directly used by SMVGEAR. The Highlight added to second paragraph of section 2.3. (Line 316-318)

Comment 4:

In discussion of method error, you show that EBI has a tolerable level of error on average across the area measured. More detail would be helpful. The biggest concern when introducing a new solver is the propagation of error over time. Here, your runs only go for 24 hours. It would be helpful to justify why you chose that run length. Furthermore, it would be helpful to include some discussion of how the error changes over time. Does the error increase as the run goes on, or is it relatively stable?

Response: Thanks for the comment. In the performance assessment run, totally 3 days simulation has been done while only the first day results were selected to present in the manuscript. The reason for this is we want to inspect the maximum extent of discrepancy of EBI, in comparison to SMVGEAR results, and whether it is acceptable. It was found the error of EBI scheme reduced as flow time increased, and gradually diminished to the range within less than 1% by the end of day 3. The error of EBI is mainly caused by the intrinsic nature of Eulerian backward (fully implicit) method, which generally produces a slower dynamic response in comparison to the real

solution.

Here is a simple example of the solution predicted by EBI and the real solution (SMVGEAR), of the two first order problems:

(1) $dC/dt = -kC$, $k=10.0$, $C_0=1.0$; (2) $dC/dt = P - kC$, $k=10.0$, $P=30.0$, and $C_0=1.0$.

It can be seen in the dynamic stage, the solution by EBI algorithm is slower, in both natural decay and external production term cases. Prediction from EBI approach to the real solution as the t increases but steady state values of these two are equal. As EBI results are always slower, in the decaying case the EBI predictions are higher and vice versa for the climbing case.

[Figure]

The initial condition used for the test run was a real concentration field derived from a normal CMAQ run with seven days spin up period rather than a clean default IC, the positively tagged species (NO_X1-NO_X6, $NO_2\_X1$-$NO_2\_X6$…) are effectively zero, but with anthropogenic emission while the null tagged species (NO_X0, $NO_2\_X0$…) are orders of magnitude higher but have no emission source. The evolution of the former is an accumulation process, and the later is a natural decay process. From the Fig 4 it can be seen that the concentration of null-tagged (IC/BC column) of EBI are higher than the SMVGEAR results (underneath the line of symmetry, especially for NO and $NO_2$), and the positively tagged species of EBI are lower. This is consistent with the above simple case illustration, and the deviation in total concentrations of are inconspicuous as the higher null-tagged concentration of EBI prediction compensates

for the lower positive-tagged results. As the flow time increases, the concentrations of positive-tagged and null-tagged species will reach a new stable state, that is oscillating within a certain bound corresponding to the emission intensities, the errors will eventually diminish to a very small level (most of them are less than 1%). Revisions made by appending additional sentences to the end of the third paragraph section 3.2, also additional data regarding the error on the second and third day has been inserted into Table 3 to show the decaying trend of error in EBI.

Comment 5: typo, should be fails to converge

Response: Thanks for the comment. The words"fail to converge" changed to "fails to converge".  (Line 130)

Comment 6: built-in

Response: Thanks for the comment. The words "build-in" has been changed to "built-in".  (Line 139)

Comment 7: Line 142-145: Need to rewrite this sentence, it is missing some words and there are grammatical issues.

Response: Thanks for the suggestion. The sentence has been modified to: "The method for improving the efficiency of the source-oriented mechanism through simplification of reaction representation and modification of the EBI ODE solver for source-oriented nitrogen species is described in Section 2. Section 3 details the testing of the improved mechanism and the source-oriented EBI solver." (Line 157–161)

Comment 8: Line 193: Period instead of semicolon

Response: Thanks for the comment. The punctuation "semicolon" has been changed to "period". (Line 212)

Comment 9: Line 279-282: Rephrase, confusing sentence.

Response: Thanks for the comment. The wording should be a little clearer and the sentence has been rewritten to "As the primary goal of this paper is to evaluate the

efficiency of the gas phase algorithm, aerosol results despite being enabled in the simulations along with cloud processes, are not included in the analyses described below.". (Line 302-304)

Comment 10: Table 1:# of species and # of reactions are rows without values

Response: Thanks for the comment. The number of species and reactions have been added into Table 1. (Line 330)

Comment 11: Section 2.2.2 Successive under-relaxation. Has this phenomenon been observed in prior EBI papers, or is this a new phenomenon. Additionally, did you come up with the relaxation coefficient or was that defined previously as well. How does it effect the time dependent concentration of species that are not at equilibrium?

Response: Thanks for the comment. The under-relaxation method is commonly used in the computational fluid dynamics (CFD) calculations to eliminate the inherent numerical instability, and the for fixed α value 0.8 is commonly used in commercial CFD codes, for scalars such as concentration or energy, and this study adopts such a concept. The successive under-relaxation is introduced for the first time in this study, to achieve quicker convergence for source-tagged mechanisms by variant α value to suit the different iteration stages, and it has not been reported in prior EBI papers. From the printout of solution versus iterations, the predicted solutions without under-relaxation showed a two-phase behavior: a rapid shift towards the true solution's proximity in the initial iterations, followed by oscillations around true solution before convergence, and fixed-point iteration frequently occurs due to numerical instability in source-tagged case. Therefore, in the very first steps larger α is favorable as it shoves the predictive concentration rapidly to true solution and subsequent smaller α is used to damp out the overshoots and avoid numerical instability. Performance with the optimal fixed α remains inferior to the dynamic α scheme due to the absence of an initial thrust. The time-dependent concentration of species is independent of the chosen α, which solely affects the number of iterations required for convergence, not the final converged solutions, whether the simulation is

time-dependent or at equilibrium. Also, the statement in 2.2.2 and Eq.11 are apparently wrong, solution for each iteration is updated with a weighted average of predicted value and previous iteration, not by values of current and previous time-step. Thus, a declaration has been inserted to the second paragraph of section 2.2.2. Corrections were made to errors in the first paragraph of section 2.2.2 and Eq. 11. (Line 279-285) Furthermore, a more detailed explanation of the successive under-relaxation method was included in an additional paragraph at the end of section 3.1. (Line 398-409)

Comment 12: Table 3: Is there a time-dependent element to the errors? Do they increase over the course of the run. One concern that is not yet addressed is whether or not this method is stable for longer runtimes, and it is not clear from the results whether the error increases over time.

Response: Thanks for the comment. As the concentration field is stable in correspondent to the external inputs (emission intensity etc.) the error will eventually be within a range. Revisions made at the end of paragraph 3, section 3.2. Also, error results for day 2 & 3 are added to Table3. "For all these species, the maximum normalized error among all grid cells is less than 15% and the mean normalized error does not exceed 4% on the first day. Subsequently, the error gradually decays over the following two days, reaching an order of magnitude of 0.1% to 1% by the third day. This indicates that the accuracy of the source-oriented EBI scheme is acceptable, as the errors are anticipated to diminish further with increasing flow time." (Line 426-431) and Table3. (Line 432)

Comment 13: Figure 3: Include a dotted line for the 1:1 matching as you have in Figure 4.

Response: Thanks for the comment. The 1:1 baseline has been added to each sub-plot of Figure 4.

Comment 14: Suggestion for Figures 3 and 4: stating in the caption which method is

baseline and which is new, switching the axis.

Response: Thanks for the suggestion. Adding a description to the legend whether old or new would aid reading. Plots with the new method as the vertical axis are more in line with common sense."New" and "baseline" have been added to the caption of Figures 3 and 4, and axis have been changed to EBI as the Y axis and SVMVGEAR as the X axis.

Comment 15: All CMAQ results figures: please include a brief description of the run in the caption. For example: Results from a one-day CMAQ simulation with XX resolution and meteorological inputs from WRF. I know that these details are given in section 2.3, but from looking at the figures alone, key details are missed.

Response: Thanks for the comment. In all CMAQ result plots, brief description of the run has been added in the caption.

Comment 16: Figures 6 and 7: Consider showing the difference between gear and ebi in a third column, as it is difficult to quantify the differences visually.

Response: Thanks for the comment. A third column has been added to Figures 6 and 7 to show the difference between GEAR and EBI.

Comment 17: SMVEAR typo.

Response: Thanks for the comment, there is a spelling error here that needs to be corrected. The word "SMVEAR" has been changed to "SMVGEAR". (Line 484)

Comment 18: Line 449-451: sentence should be rewritten for clarity.

Response: Thanks for the comment and sorry for the mistake. It is confusing that 'Test cases based on the Texas Air Quality Study 2006' appears in the conclusions for the first time. This phrase, discussing reported study results, was included in error. And it has been moved into introduction part with reference. (Line 104-108)

Comment 19: Line 456: what are "adverse meteorological conditions"? Consider

rephrasing.

Response: Thanks for the comment, adverse meteorological conditions means weather patterns that exacerbate pollution, like the stable inversions that trap winter fine PMs or the hot, sunny conditions that enhance summer ozone production. The sentence has been modified to: "under meteorological conditions that exacerbate pollution". (Line 520-521)

---

## Author Response (AR3)

**Manuscript number: EGUSPHERE-2025-44**

**Title: Improving the computation efficiency of a source-oriented chemical mechanism for the simultaneous source apportionment of ozone and secondary particulate pollutants**

Editorial office:

Figures 6, 7 may contain a territory that is disputed according to the United Nations. If and when the manuscript is accepted for final revised publication, you will be asked to choose one of the following options: (a) you could remove the disputed territory from the maps and submit new figure files, or (b) we could add a statement that some figures contain disputed territories.

Response: Thanks for the comment. After careful consideration, we choose the option (b) by adding a statement that some figures contain disputed territories.